

# FAIR v1.1: A simple emissions-based impulse response and carbon cycle model

Christopher J. Smith[1], Piers M. Forster[1], Myles Allen[2], Nicholas Leach[2], Richard J. Millar[3,4], Giovanni A. Passerello[1], and Leighton A. Regayre[1]

[1]School of Earth and Environment, University of Leeds, Leeds, UK
[2]Atmospheric Physics Department, University of Oxford, Oxford, UK
[3]College of Engineering, Mathematics and Physical Sciences, University of Exeter, Exeter, UK
[4]Environmental Change Institute, University of Oxford, Oxford, UK

*Correspondence to:* C.J.Smith (C.J.Smith1@leeds.ac.uk)

**Abstract.** Simple climate models can be valuable if they are able to replicate aspects of complex fully coupled earth system models. Larger ensembles can be produced, enabling a probabilistic view of future climate change. A simple emissions-based climate model, FAIR, is presented which calculates atmospheric concentrations of greenhouse gases and effective radiative forcing (ERF) from greenhouse gases, aerosols, ozone precursors and other agents. The ERFs are integrated into global mean

surface temperature change. Model runs are constrained to observed temperature change from 1880 to 2016 and produce a range of future projections under the Representative Concentration Pathway (RCP) scenarios. For the historical period the ERF time series in FAIR emulates the results in the IPCC Fifth Assessment Report (AR5), whereas for RCP historical and future scenarios, the greenhouse gas concentrations in FAIR closely track the observations and projections in the RCPs. The constrained estimates of equilibrium climate sensitivity (ECS) of 2.79 (1.97 to 4.08) K, transient climate response (TCR) of 1.47

(1.03 to 2.23) K and transient climate response to cumulative $CO_2$ emissions (TCRE) of 1.43 (1.01 to 2.16) K (1000 GtC)$^{-1}$ (median and 5–95% credible intervals) are in good agreement, with tighter uncertainty bounds, than AR5 (1.5 to 4.5 K, 1.0 to 2.5 K, and 0.8 to 2.5 K respectively). The ranges of future projections of temperature and ranges of estimates of ECS, TCR and TCRE are moderately sensitive to the historical temperature dataset used to constrain, prior distributions of ECS/TCR parameters, aerosol radiative forcing relationship and ERF from a doubling of $CO_2$. Taking these sensitivities into account,

there is no evidence to suggest that the median and credible range of observationally constrained TCR or ECS differ from climate model-derived estimates. However, the range of temperature projections under the RCP scenarios for 2081–2100 in the constrained FAIR model ensemble are lower than the emissions-based estimates reported in AR5.





## 1 Introduction

Most multi-model studies, such as the Coupled Model Intercomparison Project (CMIP) which produces headline climate projections for the Intergovernmental Panel on Climate Change (IPCC) assessment reports, compare atmosphere-ocean general circulation models that are run with prescribed concentrations of greenhouse gases. Greenhouse gas and aerosol emissions

time series are provided by integrated assessment modelling groups based on socio-economic narratives (Moss et al., 2010; Meinshausen et al., 2011b), which are then converted to atmospheric concentrations by simple models such as MAGICC6 (Meinshausen et al., 2011a). On the other hand, earth system models can be run in emissions mode, where emissions of carbon dioxide are used as a starting point and the atmospheric $CO_2$ concentrations are calculated interactively in the model. The earth system components of land and ocean carbon fluxes are based on biogeochemical relationships, with atmospheric

concentration changes being the residual of emissions minus absorption by land and ocean sinks. While many models include the functionality to be run in $CO_2$ emissions mode, these integrations were not the main focus of CMIP5 (the fifth phase of CMIP; Taylor et al. (2012)).

Earth system models in CMIP5 all show a positive carbon-cycle feedback, meaning that as surface temperature increases, land and carbon sinks become less effective at absorbing $CO_2$ and a larger proportion of any further emitted carbon will remain

in the atmosphere (Friedlingstein, 2015). The various feedback strengths are nevertheless very model dependent (Friedlingstein et al., 2006). The causes of model diversity in feedback strengths was explored only superficially in CMIP5 owing to the number of earth system models available and the experimental design (Taylor et al., 2012; Millar et al., 2017b). Only $CO_2$-emission driven experiments under historical and RCP8.5 forcing compared to pre-industrial control emissions were core experiments in CMIP5 for emissions-driven earth system models (Taylor et al., 2012). While $CO_2$ is the most important climate forcer,

individual models may also respond differently to other emissions. These forcings can also introduce uncertainty that is not captured in concentration-driven or $CO_2$-only driven model experiments (Matthews and Zickfeld, 2012; Tachiiri et al., 2015). As non-$CO_2$ forcing impacts temperature which affects the efficiency of carbon sinks, non-$CO_2$ forcing agents themselves influence the carbon cycle.

Simple models can be used to emulate radiative forcing and temperature responses to emissions and atmospheric concentra-

tions, and can be tuned to replicate the behaviour of individual climate and earth system models (Meinshausen et al., 2011a; Good et al., 2011, 2013; Geoffroy et al., 2013). In the IPCC Fifth Assessment Report (AR5), a simple carbon-cycle model was suggested, calibrated to present-day conditions (Supplementary Material to Myhre et al., 2013b). This model used fixed time constants for the decay of atmospheric $CO_2$ with no feedbacks assumed for biospheric carbon uptake or temperature. It was introduced for the purposes of calculating global warming potentials, built on the impulse response model of Boucher and

Reddy (2008). Millar et al. (2017b) showed that this AR5 model does not sufficiently capture the time-evolving dependency of carbon sinks with cumulative carbon emissions and temperatures. They introduced the Finite Amplitude Impulse Response (FAIR) model (version 1.0) that tracks the time-integrated airborne fraction of carbon and uses this to determine the efficiency of carbon sinks. This model, version 1.1, is developed further in this paper.



Simple models themselves span a range of complexity scales. At one end are one-box and multi-box impulse response models (Boucher and Reddy, 2008; Myhre et al., 2013b; Geoffroy et al., 2013), and at the other are models that treat land and ocean processes separately, with sophisticated carbon cycles, state-varying atmospheric sinks and ocean upwelling-diffusion modelling (e.g. MAGICC6, Meinshausen et al. (2011a)). Providing that simple models can adequately capture the key features of more complex climate and earth system models, they can be used to create ensemble estimates of future scenarios that

would not be possible due to computational demands, and provide the capacity for probabilistic interpretations of future climate change.

FAIR v1.0 is a simple $CO_2$ emissions-based carbon cycle model, written in Python 2, that is well-validated to the response of earth system models. FAIR v1.1 is extended to calculate non-$CO_2$ greenhouse gas concentrations from emissions, aerosol forcing from aerosol precursor emissions, tropospheric and stratospheric ozone forcing from the emissions of precursors, and

forcings from black carbon on snow, stratospheric methane oxidation to water vapour, contrails and land use change. Although many of these forcings exhibit substantial regional variations, this is expected to be smoothed out in the global mean. Forcings from volcanic eruptions and solar irradiance fluctuations are supplied externally. The extension to non-$CO_2$ emissions makes FAIR v1.1 applicable for assessing scenarios with a broad range of emissions commitments.

This paper introduces the FAIR model in section 2, including the key changes from versions 1.0 to 1.1. Section 3 then

discusses the generation of a large ensemble of input parameters to the FAIR model which is run and results described in section 4. A sensitivity analysis to some of the key inputs to the large ensemble is given in section 5. Section 6 provides a summary.

## 2  Development of FAIR v1.1 and differences to v1.0

FAIR v1.1 takes emissions of greenhouse gases, short-lived climate forcers, the fraction of total nitrogen oxides (NOx) emitted

by the aviation sector, fraction of total methane attributable to fossil fuels, and natural forcing from solar variability and volcanoes, as its inputs. The atmospheric concentrations of greenhouse gases are calculated from new emissions minus the decay of the current atmospheric burden, which is determined by the atmospheric lifetime of each gas. For $CO_2$, atmospheric concentrations are calculated from a simple representation of the carbon cycle which includes temperature and saturation dependency of land and ocean sinks and includes a proportion of methane oxidised to $CO_2$. The effective radiative forcing

(ERF) from 13 different forcing agents (table 1) is determined from the concentrations of each greenhouse gas, plus emissions of short-lived climate forcers and natural forcing. From the ERF, temperature change is calculated. The change in temperature feeds back into the carbon cycle, which impacts the atmospheric lifetime of carbon dioxide. A flow diagram outlining the key processes is provided in fig. 1.





## 2.1 Emissions to concentrations

### 2.1.1 Carbon dioxide and carbon cycle

The carbon cycle component in FAIR v1.0 is described in detail by Millar et al. (2017b) and an overview is provided here. The FAIR model uses anthropogenic fossil and land-use $CO_2$ emissions as input and partitions them into four boxes $R_i$ (with partition fraction $a_i$ and $\sum_0^3 a_i = 1$) representing the differing time scales of carbon uptake by geological processes ($\tau_0$), the deep ocean ($\tau_1$), biosphere ($\tau_2$) and ocean mixed layer ($\tau_3$). The atmospheric concentrations of $CO_2$ and its relationship to each box is

$$C_{CO_2} = C_{CO_2,pi} + \sum_{i=0}^{3} \frac{R_i}{M_a} \frac{w_{CO_2}}{w_a} \tag{1}$$

with $C_{CO_2,pi}$ equal to 278 ppm and pi representing pre-industrial. $M_a = 5.1352 \times 10^{18}$ kg is the dry mass of the atmosphere

and $w_{CO_2}$ and $w_a$ are the molecular weights of $CO_2$ and dry air. The governing equations for the four boxes are

$$\frac{dR_i}{dt} = a_i E_{CO_2} - \frac{R_i}{\alpha \tau_i}; \qquad i = 0, \ldots, 3 \tag{2}$$

with $E_{CO_2}$ the emissions of $CO_2$.

The four time constants $\tau_i$ are scaled by a factor $\alpha$ depending on the 100-year integrated impulse response function (iIRF$_{100}$), which represents the 100-year average airborne fraction of a pulse of $CO_2$ (Joos et al., 2013). $\alpha$ is found by equating two

different expressions for iIRF$_{100}$

$$\sum_{i=0}^{3} \alpha a_i \tau_i \left[ 1 - \exp\left(\frac{-100}{\alpha \tau_i}\right) \right] = r_0 + r_C C_{acc} + r_T T \tag{3}$$

and finding the unique root $\alpha$ (Millar et al., 2017b). The right hand side of eq. (3) proposed by Millar et al. (2017b) is a simplified expression for iIRF$_{100}$ that depends on the total accumulated carbon in land and ocean sinks $C_{acc} = (\sum_t E_{CO_2,t}) - (C_{CO_2} - C_{CO_2,pi})$ and temperature change $T$ since the pre-industrial era that simulates the behaviour of earth system models

well. This increase in iIRF$_{100}$ and scaling of the time constants by $\alpha$ accounts for the land and ocean carbon sinks approaching saturation as more carbon is added to them ($r_C$ parameter). In earth system models it is also observed that the efficiency of carbon sinks decreases with increasing temperature ($r_T$ parameter; Fung et al. (2005); Friedlingstein et al. (2006)). Following Millar et al. (2017b) we use $r_C = 0.019$ yr GtC$^{-1}$ and $r_T = 4.165$ yr K$^{-1}$, but in contrast to Millar et al. (2017b) a pre-industrial $r_0 = 35$ years is used rather than their 32.4 years. This facilitates better agreement with present-day $CO_2$ atmospheric

concentrations when spun up from 1765 with historical $CO_2$ and non-$CO_2$ emissions. This parameter combination is consistent with a present-day iIRF$_{100}$ diagnosed from more complex carbon-cycle models (Joos et al., 2013) with a reference $CO_2$ concentration of 389 ppm.



### 2.1.2 Other greenhouse gases

A one-box decay model is assumed for other greenhouse gases where the sink is an exponential decay of the existing gas concentration. New emissions are converted to concentrations in year $t$ by

$$\delta C_t = \frac{E_t}{M_a}\frac{w_a}{w_f} \tag{4}$$

where $E_t$ is the emissions of gas in year $t$, $M_a = 5.1352 \times 10^{18}$ kg is the mass of the atmosphere and $w_f$ is the molecular mass of the greenhouse gas. The model updates the atmospheric concentrations at year $t$ based on new emissions and the natural sink by

$$C_t = C_{t-1} + \frac{1}{2}(\delta C_{t-1} + \delta C_t) - C_{t-1}(1 - \exp(-1/\tau)) \tag{5}$$

where $\tau$ is the atmospheric lifetime of each gas (table 2).

For $CH_4$ and $N_2O$, time-varying natural emissions are included in $E_t$ (fig. 2) in order to closely match the atmospheric concentrations of these gases in Meinshausen et al. (2011b), including the 1765 concentrations when the 1765 natural emissions are run to steady state. The time series of natural emissions used include abrupt step changes in places in order to counterbalance differing rates of change in anthropogenic emissions, for example a rapid increase in anthropogenic $N_2O$ emissions in the 1940s. In 2011 the natural emissions of $CH_4$ and $N_2O$ are taken as 195 Mt $CH_4$ yr$^{-1}$ and 8.75 Mt $N_2$-eq yr$^{-1}$ which are close to the best-estimate present-day emissions of 202 Mt $CH_4$ yr$^{-1}$ and 9.1 Mt $N_2$-eq yr$^{-1}$ (Prather et al., 2012).

Natural emissions of $CO_2$ are not included as the carbon cycle model is more complex than the single box used for other gases and it is assumed that natural sources and natural sinks are in balance. For other greenhouse gases, natural emissions are assumed to be zero except for $CF_4$, $CH_3Br$ and $CH_3Cl$. Rather than using the pre-industrial emissions for these three species in Meinshausen et al. (2011b), we instead match the pre-industrial concentrations in Meinshausen et al. (2011b) by running eq. (5) to steady state with the atmospheric lifetimes from table 2. We obtain background natural emissions of 0.0109 kt $CF_4$ yr$^{-1}$, 69.7 kt $CH_3Br$ yr$^{-1}$ and 2716 kt $CH_3Cl$ yr$^{-1}$. The Meinshausen et al. (2011b) emissions are then adjusted by adding on these background values of these three species and subtracting the Meinshausen et al. (2011b) year-1765 emissions from all years. In total, 31 greenhouse gas species are used (table 2). Other than $CO_2$, $CH_4$ and $N_2O$, the remaining gases can be subdivided into those covered by the Kyoto Protocol (HFCs, PFCs, $SF_6$), and the ozone depleting substances (ODSs) covered by the Montreal Protocol (CFCs, HCFCs and other chlorinated and brominated compounds).

The best estimate of $\tau$ for each gas except methane is used from AR5 (Myhre et al., 2013b, table 8.A.1). We find that using a constant methane lifetime of 9.3 years with small variations to the natural emissions provides a time series of historical atmospheric methane concentrations that differs from Meinshausen et al. (2011b) by at most 1.2% and usually much less. The methane lifetime of 9.3 years used is significantly lower than the 12.4 years in AR5. This latter figure includes the feedback of methane emissions on its own lifetime (a factor of 1.34; Holmes et al. (2013), also used in AR5) and is used for perturbation calculations against a constant background concentration. Dividing the perturbation lifetime by 1.34 gives the atmospheric burden lifetime of 9.3 years.





### 2.1.3 Methane oxidation to CO₂

The oxidation of $CH_4$ produces additional $CO_2$ if it is of fossil origin, and this has also been accounted for. Methane is assumed to be from fossil sources if it is of anthropogenic origin and arises from the transport, energy or industry sectors. We use the breakdown of emissions by sector from the RCP Database (2009) . A best estimate of 61% of the methane lost through reaction

with the hydroxyl radical in the troposphere (the dominant loss pathway) is converted to $CO_2$ (Boucher et al., 2009). This is treated as additional emissions of $CO_2$:

$$E_{\mathrm{CH_4 \to CO_2}} = 0.61 f_{\mathrm{CH_4 fos}}(C_{\mathrm{CH_4}} - C_{\mathrm{CH_4,pi}})(1 - \exp(-1/\tau_{\mathrm{CH_4}})) \tag{6}$$

where $f_{\mathrm{CH_4 fos}}$ is the fraction of anthropogenic methane attributable to fossil sources and $\tau_{\mathrm{CH_4}}$ is 9.3 years.

Oxidation of CO and non-methane volatile organic compounds (NMVOCs) to $CO_2$ is not included as to not double count the carbon that is included in national $CO_2$ emissions inventories (Daniel and Solomon, 1998; Gillenwater, 2008).

## 2.2 Effective radiative forcing

The ERF from 13 different forcing agent groups are considered: $CO_2$, $CH_4$, $N_2O$, other greenhouse gases, tropospheric $O_3$, stratospheric $O_3$, stratospheric water vapour, contrails, aerosols, black carbon on snow, land use change, solar irradiance and volcanoes (table 1). ERF, which accounts for all (stratospheric plus tropospheric) rapid adjustments, corresponds better to temperature change than "traditional" stratospherically adjusted radiative forcing (RF) (Myhre et al., 2013b; Forster et al., 2016). Therefore, we use relationships for ERF where they exist.

### 2.2.1 Carbon dioxide, methane and nitrous oxide

We use the updated Etminan et al. (2016) RF relationships for $CO_2$, $CH_4$ and $N_2O$, which for the first time includes band overlaps between $CO_2$ and $N_2O$. It also includes a significant upward revision of the $CH_4$ RF due to inclusion of previously neglected shortwave absorption, compared to the previous relationships of Myhre et al. (1998) used in AR5. Although Etminan et al. (2016) calculate RF, Myhre et al. (2013b) showed that over the industrial era ERF agrees with RF for these three gases,

although with a doubled uncertainty range. The Etminan et al. (2016) relationships are reproduced in eqs. (7) to (9), where $C$ (ppm), $M$ and $N$ (ppb) have been used to represent concentrations of $CO_2$, $CH_4$ and $N_2O$, and the subscript pi representing pre-industrial concentrations.

$$F_{\mathrm{CO_2}} = \left[(-2.4 \times 10^{-7})(C - C_{\mathrm{pi}})^2 + (7.2 \times 10^{-4})|C - C_{\mathrm{pi}}| - (1.05 \times 10^{-4})(N + N_{\mathrm{pi}}) + 5.36\right] \times \log\left(\frac{C}{C_{\mathrm{pi}}}\right) \tag{7}$$

$$F_{\mathrm{N_2O}} = \left[(-4.0 \times 10^{-6})(C + C_{\mathrm{pi}}) + (2.1 \times 10^{-6})(N + N_{\mathrm{pi}}) - (2.45 \times 10^{-6})(M + M_{\mathrm{pi}}) + 0.117\right] \times \left(\sqrt{N} - \sqrt{N_{\mathrm{pi}}}\right) \tag{8}$$

$$F_{\mathrm{CH_4}} = \left[(-6.5 \times 10^{-7})(M + M_{\mathrm{pi}}) - (4.1 \times 10^{-6})(N + N_{\mathrm{pi}}) + 0.043\right] \times \left(\sqrt{M} - \sqrt{M_{\mathrm{pi}}}\right). \tag{9}$$



### 2.2.2 Other well-mixed greenhouse gases

For all greenhouse gases in table 2 except $CO_2$, $CH_4$ and $N_2O$ the ERF is assumed to be a linear relationship of the change in gas concentration $C_i$ since the pre-industrial era by its radiative efficiency $\eta_i$ [W m$^{-2}$ ppb]:

$$F_i = \eta_i(C_i - C_{i,\mathrm{pi}}); \qquad i \in \{\text{minor greenhouse gases}\}. \tag{10}$$

where radiative efficiencies are given in table 2 and $C_i$ are converted to ppb.

### 2.2.3 Tropospheric ozone

Tropospheric ozone is formed from a complex chemical reaction chain from emissions of $CH_4$, NOx, CO and NMVOC. Furthermore its concentration is more variable in space and time than for the well-mixed greenhouse gases. Therefore, we do not calculate a globally averaged concentration. We use a multivariate linear regression between the emissions of $CH_4$, NOx,

CO and NMVOC (predictors) and tropospheric ozone ERF diagnosed in AR5 such that

$$F_{O_3\mathrm{tr}} = \beta_{CH_4}E_{CH_4} + \beta_{NOx}E_{NOx} + \beta_{CO}E_{CO} + \beta_{NMVOC}E_{NMVOC} \tag{11}$$

Only anthropogenic $CH_4$ emissions are considered in eq. (11), defining tropospheric ozone changes in FAIR as a purely anthropogenic forcing. The regression approach has the effect of isolating the effect of each precursor species on its total contribution to the tropospheric ozone forcing. The regression coefficients $\beta_{CH_4} = 2.82 \times 10^{-4}$, $\beta_{NOx} = 99.78 \times 10^{-4}$, $\beta_{CO} = 1.07 \times 10^{-4}$ and $\beta_{NMVOC} = -9.36 \times 10^{-4}$ W m$^{-2}$ (Mt yr$^{-1}$)$^{-1}$. Unlike Stevenson et al. (2013) we find a negative correspondence on ozone

forcing with emissions of NMVOC with this simplified method. All other emissions species result in a positive tropospheric ozone forcing.

### 2.2.4 Stratospheric ozone

The stratospheric ozone ERF is calculated using the functional relationship in Meinshausen et al. (2011a), namely

$$F_{O_3\mathrm{st}} = a(bs)^c. \tag{12}$$

$a = -1.46 \times 10^{-5}$, $b = 2.05 \times 10^{-3}$ and $c = 1.03$ in eq. (12) are fitting parameters that are found by a least-squares curve fit between eq. (12) and the stratospheric ozone ERF timeseries from AR5. $s$ is the equivalent effective stratospheric chlorine (EESC) from all ozone depleting substances, calculated as (Newman et al., 2007)

$$s = r_{CFC11} \sum_{i \in ODS} \left( n_{Cl}(i)C_{i,t-3}\frac{r_i}{r_{CFC11}} + 45n_{Br}(i)C_{i,t-3}\frac{r_i}{r_{CFC11}} \right). \tag{13}$$

$r_i$ represents fractional release values for each ODS compound taken from Daniel and Velders (2011) and reproduced in table 2.

$n_{Cl}$ and $n_{Br}$ represent the number of chlorine and bromine atoms in compound $i$ with the factor of 45 in eq. (13) indicating that bromine is 45 times more effective at stratospheric ozone depletion than chlorine (Daniel et al., 1999). The concentrations $C_i$ are expressed in ppb and are taken from year $t - 3$, representing the time for the air to be transported to the stratosphere.



### 2.2.5 Stratospheric water vapour from methane oxidation

In AR5, the ERF from the stratospheric water vapour oxidation of methane was assumed to be 15% of the methane ERF. This approach is also taken in FAIR.

### 2.2.6 Contrails

Meinshausen et al. (2011b) did not include a forcing timeseries for contrails or contrail-induced cirrus, which contribute a small positive ERF (Boucher et al., 2013). It is assumed that contrail ERF is proportional to the level of air traffic, which is in turn is proportional to aircraft emissions (Lee et al., 2009). We use aviation NOx emissions for this purpose, which are obtained from the RCP database. The ERF from contrails $F_{\mathrm{con}}$ is scaled by the ratio of aircraft NOx emissions in a given year compared to 2011 and multiplied by the 2011 ERF:

$$F_{\mathrm{con}} = \frac{E_{\mathrm{NOx,avi}}}{E_{\mathrm{NOx,avi,2011}}} F_{\mathrm{con},2011}. \tag{14}$$

This gives a coefficient of $F_{\mathrm{con},2011}/E_{\mathrm{NOx,avi,2011}} = 0.0152$ W m$^{-2}$ (Mt-aviNOx yr$^{-1}$)$^{-1}$.

### 2.2.7 Aerosols

The radiatively active aerosols in the RCP datasets are fossil fuel black carbon (BC), fossil fuel organic carbon (OC), sulfate, nitrate, biomass aerosol and mineral dust. Aerosols have a lifetime of the order of days (Kristiansen et al., 2016), and the emissions are converted to forcing without an intermediate concentration step.

The aerosol ERF contains contributions from aerosol-radiation interactions (ERFari) and from aerosol-cloud interactions (ERFaci). ERFari includes the direct radiative effect of aerosols, in addition to rapid adjustments due to changes in the atmospheric temperature, humidity and cloud profile (formerly the semi-direct effect; Boucher et al. (2013)). In the AR5 ERF time series, the components of aerosol forcing are not considered separately, but the year 2011 best estimates of $-0.45$ W m$^{-2}$ for ERFari and $-0.45$ W m$^{-2}$ ERFaci are given.

We use a multi-linear regression approach to isolate the effect of each emitted species (BC, OC, SOx, NOx and NH$_3$) on the total aerosol forcing by regressing the AR5 historical time series against the emissions in Meinshausen et al. (2011b):

$$F_{\mathrm{aero}} = \gamma_{\mathrm{BC}}E_{\mathrm{BC}} + \gamma_{\mathrm{OC}}E_{\mathrm{OC}} + \gamma_{\mathrm{SOx}}E_{\mathrm{SOx}} + \gamma_{\mathrm{NOx}}E_{\mathrm{NOx}} + \gamma_{\mathrm{NH_3}}E_{\mathrm{NH_3}}. \tag{15}$$

We assume emitted BC and OC correspond directly to BC and OC forcing, and that SOx corresponds directly to sulfate forcing. Following Shindell et al. (2009) we assume a 60% contribution to nitrate aerosol forcing from NH$_3$ and 40% from NOx. Biomass burning aerosol has a net zero forcing in 2011 and is ignored, and mineral dust, which does not scale directly with an emitted component, is also disregarded.

To provide bounds on the contribution of each species to the total aerosol forcing, we use the maximum and minimum of the multi-model radiative forcing results from Aerocom (Myhre et al., 2013a) for the direct forcing of each species. The sum of direct forcing is $-0.31$ W m$^{-2}$ for the best estimate from Aerocom models, so the upper bounds (i.e. furthest bound from



zero) of each species is scaled by $-0.9/-0.31$ to account for the rapid adjustments in ERFari and ERFaci that contribute to the total aerosol forcing. This leads to the estimates of 2011 ERF and regression coefficients in table 3.

In this simplified treatment the ERF in 2011 for BC and sulfate are towards the weaker end of the allowable range (the range guided by Aerocom models, scaled for adjustments) while the ERFs from nitrate aerosol precursors are at the most extreme strong end. We also assume that the relationship between aerosol ERF and emissions is linear, and do not discriminate on the strength of adjustments by species. There is some evidence to suggest that the aerosol forcing may increase as a sub-linear, potentially logarithmic, function of emissions as the increase in cloud albedo, one component of ERFaci, saturates with

increasing aerosol emissions (Twomey, 1991; Carslaw et al., 2013). An alternative relationship is explored in the sensitivity analysis.

### 2.2.8    Black carbon on snow

The best-estimate ERF of 0.04 W m$^{-2}$ in AR5 for 2011 is compared to the BC emissions in 2011 from Meinshausen et al. (2011b), with this scaling factor assumed to hold for all years. The relationship is given by

$$F_{\text{BCsnow}} = 0.00494 E_{\text{BC}}, \tag{16}$$

where $E_{\text{BC}}$ is BC emissions in Mt yr$^{-1}$.

### 2.2.9    Land use change

Land use forcing is largely driven by surface albedo change (Andrews et al., 2017), which is often due to deforestation for agriculture (Myhre and Myhre, 2003). Cropland has a higher albedo than the forest that it replaces, reflecting more incident solar radiation and therefore resulting in a negative ERF. Deforestation produces land-use related CO$_2$ emissions. The total

amount of deforestation since pre-industrial times could therefore be expected to scale with cumulative land-use related CO$_2$ emissions. This approach is taken in FAIR. A regression of non-fossil CO$_2$ emissions against land use ERF in AR5 gives

$$F_{\text{landuse}} = 1.14 \times 10^{-3} \sum_{j=0}^{t} E_{\text{CO}_2\text{land},j}, \tag{17}$$

where the coefficient has units W m$^{-2}$ (Gt C)$^{-1}$.

### 2.2.10    Solar variability

The SOLARIS-HEPPA v3.2 solar irradiance dataset prepared for CMIP6 is used to generate the solar ERF, which includes projections of the variation in future solar cycles from 1850 to 2300 (Matthes et al., 2017). ERF from solar forcing is calculated as the change in solar constant since 1850 divided by 4 (average insolation) and multiplied by 0.7 (representing planetary co-albedo). This approach is also used in Meinshausen et al. (2011b). Prior to 1850, we revert to the solar forcing from AR5.



### 2.2.11 Volcanic aerosol

Historical volcanic forcing is punctuated by several large eruptions that cause large but short-lived negative forcing episodes, with several smaller eruptions that cause year-to-year changes in the volcanic forcing. The ERF from volcanic eruptions in AR5 is used without modification.

### 2.3 Temperature change

In simple impulse-response models, forcing is related to total temperature change in year $t$, $T_t$, by a two-time constant model (Boucher and Reddy, 2008; Myhre et al., 2013b; Millar et al., 2015, 2017b). FAIR v1.1 takes this approach with a small modification compared to FAIR v1.0 to allow for forcing-specific efficacies $\epsilon_j$ such that

$$T_{t,i} = T_{t-1,i}\exp(1/d_i) + \sum_{j=1}^{13} (c_i\epsilon_j F_j(1 - \exp(1/d_i)); \qquad i = 1, 2. \tag{18}$$

Owing to the use of ERF rather than RF in FAIR v1.1 and its better correspondence with temperature, efficacies are assumed to be unity for all forcing agents except black carbon on snow ($j = 10$), where an efficacy of 3 is used following Bond et al. (2013). The coefficients $d_1$ and $d_2$ govern the slow ($i = 1$) and fast ($i = 2$) temperature changes from a response to radiative forcing from the upper ocean and the deep ocean respectively (Millar et al., 2015). The total temperature change in year $t$ is the sum of the slow and fast components, i.e. $T_t = T_{t,1} + T_{t,2}$. $F_j$ represents the 13 individual forcing agents in year $t$ calculated in section 2.2 (see also table 1). $d_1$ and $d_2$ default to 4.1 and 239 years which are fit to match the mean of CMIP5 models (Geoffroy et al., 2013). The coefficients $c_1$ and $c_2$ (units K W$^{-1}$ m$^2$) are determined by solving a matrix equation given TCR, ECS, $d_1$, $d_2$ and the ERF from a doubling of $CO_2$, $F_{2\times} = 3.71$ W m$^{-2}$ (Myhre et al., 2013b):

$$T_{\mathrm{ECS}} = F_{2\times}(c_1 + c_2); \tag{19}$$

$$T_{\mathrm{TCR}} = F_{2\times}\left(c_1\left(1 - \frac{d_1}{D}\left(1 - \exp\left(-\frac{D}{d_1}\right)\right)\right) + c_2\left(1 - \frac{d_2}{D}\left(1 - \exp\left(\frac{D}{d_2}\right)\right)\right)\right) \tag{20}$$

giving the relative contributions to the fast and slow components of the warming. $D = \log(2)/\log(1.01) \approx 69.7$ years is the time to a doubling of $CO_2$ with a compound 1% per year increase in $CO_2$ concentrations.

## 3 Projections using a large ensemble

To test the model response to a range of forcing pathways, we perform a 100,000-member Monte Carlo simulation using emissions from the RCP datasets (Meinshausen et al., 2011b). Emissions themselves are not altered from the RCP timeseries, but the TCR, ECS, carbon cycle response to increasing temperature ($r_T$) and cumulative emissions ($r_C$) along with the pre-industrial value of iIRF$_{100}$ ($r_0$), plus the ERF scale factors for each of the 13 forcing agents, are drawn from distributions. FAIR is run from 1765 (the start of the RCP emissions datasets) to 2100.



### 3.1 Constraint to historical temperature observations

As a wide range of forcing, and thus temperature, scenarios can be generated, there are a proportion of ensemble members generated that fall outside the range of plausibility. We constrain the full 100,000 member ensemble (hereafter FULL) to the observed temperature change from the Cowtan and Way (2014) dataset (hereafter C&W) to assess plausibility; ensemble members that satisfy the temperature constraint are designated as Not Ruled Out Yet (NROY) and the majority of the discussion of the results in section 4 focuses on this dataset. We rebase all of the temperatures to the 1861–1880 mean following Richardson

et al. (2016), to represent a "pre-industrial" state that is relatively free from volcanic eruptions but with a reasonable global coverage of temperature observations. An ordinary least-squares regression of temperature change versus time from 1880–2016 is used to calculate the linear warming trend in each ensemble member. The regression is also performed for the C&W "observational" dataset to estimate the observed warming rate. The confidence interval around the C&W warming rate is inflated by a factor that represents the lag-1 autocorrelation of residuals (i.e. the trend-line estimate from the regression minus the C&W

"observations") which accounts for internal climate variability (Santer et al., 2008; Thompson et al., 2015) and is the same method used in AR5 to estimate linear temperature trends.

The C&W observed warming from 1880–2016 is $0.95 \pm 0.17$ K, higher than the HadCRUT4 estimate of $0.91 \pm 0.18$ K for the same timeframe and the AR5 estimate of $0.85 \pm 0.20$ K for 1880–2012 (Hartmann et al., 2013). The infilling of grid boxes where no or limited data are available accounts for these differences, as sparse observations are typically in polar regions which warm faster than the global mean (Cowtan and Way, 2014).

Under this constraint approximately 26% of the FULL ensemble is retained in NROY.

### 3.2 Sampling ECS and TCR

TCR and ECS are proposed to follow a joint lognormal distribution, based on the evaluated means and standard deviations of TCR and ECS from CMIP5 models (Flato et al., 2013; Forster et al., 2013), and representative of the distributions of ECS and TCR in the literature (Meinshausen et al., 2009; Rogelj et al., 2012; Millar et al., 2017b). We sample 100,000 ECS/TCR pairs; sampled pairs where ECS < TCR are rejected and redrawn. A joint distribution is used because ECS and TCR are highly correlated and low values of the realised warming fraction (TCR divided by ECS) are inconsistent with models and

observations (Millar et al., 2015). The correlation coefficient in CMIP5 models ($r = 0.81$) is used in the construction of the joint distribution.

### 3.3 Sampling thermal response and carbon cycle parameters

We allow $F_{2\times}$, the ERF due to a doubling of $CO_2$, to assume a Gaussian distribution with 5–95% confidence interval of 20% around the best estimate ERF of 3.71 W m$^{-2}$ (Myhre et al., 2013b). $d_1$ and $d_2$ in eq. (18) are also varied based on truncated

Gaussian distributions (no values outside $\pm 3\sigma$ allowed, primarily to prevent unrealistically small or negative values of the fast response $d_2$) with mean and standard deviation equal to the CMIP5 model estimates in Geoffroy et al. (2013).



Some uncertainty in the carbon cycle parameters is assumed with samples of $r_0$, $r_C$ and $r_T$ taken from Gaussian distributions. $r_0$, $r_C$ and $r_T$ are given 5–95% confidence intervals of 13% of the default parameter value following Millar et al. (2017b).

### 3.4 Sampling ERF uncertainties

The uncertainty in each of the 13 forcing components is modelled following the 5–95% confidence intervals for each forcing from AR5 (Myhre et al., 2013b, Table 8.6) in 2011 (table 1). This is achieved by scaling the ERF values calculated in section 2.2. The scaling factor is applied to the whole time series. Most uncertainties are assumed to be Gaussian, the exceptions being contrails and BC on snow which are lognormally distributed, and aerosols which are modelled as two half-Gaussian distributions, treating values above and below the best estimate separately. These ERF uncertainties are assumed to be uncorrelated 25 with each other.

## 4 Results from the NROY ensemble for the RCP scenarios

### 4.1 ECS and TCR

The FULL and NROY joint and marginal distributions of ECS and TCR are shown in fig. 3.

The temperature constraint in NROY results in distributions of ECS and TCR that are lower than in FULL. Much of the prior 30 sample space in which ECS and TCR are larger than the AR5 likely ranges has been rejected in the NROY distribution. While the possibility that ECS > 5 K cannot be ruled out, it appears less likely than would be inferred from CMIP5 models, although it should be stressed that time-varying feedbacks are not accounted for FAIR which would allow ECS to increase over time (Armour, 2017). From the marginal distributions, we estimate that ECS and TCR are 2.79 (1.97 to 4.08) K and 1.47 (1.03 to 2.23) K (median; (5–95% range)) respectively in the NROY ensemble, similar to but a little more tightly constrained than the AR5 ranges of 1.5 to 4.5 K and 1.0 to 2.5 K. The ratio of TCR to ECS, the realised warming fraction (RWF), is approximately 5 independent of TCR in CMIP5 models (Millar et al., 2015) and the prior distribution could alternatively be defined in terms of the TCR and RWF joint distribution, which is explored in section 5. The FULL and NROY median and 5 to 95% ranges of RWF are 0.56 (0.41 to 0.76) and 0.53 (0.40 to 0.71) respectively, which is in the range of CMIP5 models (0.45 to 0.75, Millar et al. (2017b)).

### 4.2 Historical and future greenhouse gas concentrations

The historical (1765–2005) greenhouse gas concentrations from the RCP scenarios in Meinshausen et al. (2011b) were assimilated from observations of in-situ and ice core records and represent a best estimate of the actual concentrations over this period. We therefore assume that the RCP database represents the best estimate of the historical concentrations and compare our estimates from the NROY ensemble using the emissions-driven model. Emissions of greenhouse gases that produced these concentrations were generated from MAGICC6.



The FAIR model reproduces the historical concentrations of greenhouse gases (fig. 4). The atmospheric concentrations of $CO_2$ estimated from FAIR are up to 9 ppm lower than MAGICC6 in the period 1900–1950 (fig. 4a). A simple carbon cycle model cannot reproduce the kinks in the observational $CO_2$ trend without large changes in the input emissions. However, between 1950 and 2005, the differences between the two curves are small. The post-2005 atmospheric $CO_2$ concentrations are slightly higher than those estimated by MAGICC6 for the RCP scenarios, but the MAGICC6 concentrations are within the

uncertainty range from the NROY ensemble. FAIR projects best estimate $CO_2$ concentrations of 425 (410–440), 547 (523–571), 686 (655–719) and 964 (915–1015) ppm for RCP2.6, RCP4.5, RCP6 and RCP8.5 in 2100. Here, the uncertainty in $CO_2$ concentrations relates to the range of carbon cycle parameters and the temperature dependence on carbon uptake sampled in the large ensemble.

The historical $CH_4$ and $N_2O$ concentrations in FAIR have been tuned to be in broad agreement to Meinshausen et al.

(2011b) by adjusting natural emissions as described previously (fig. 4b,c). There are some small differences in the future $CH_4$ concentrations in the RCPs using a fixed methane lifetime and constant (present-day) natural emissions, but future $N_2O$ concentration projections are very similar to Meinshausen et al. (2011b).

Kyoto Protocol gases have been grouped as HFC134a-eq based on their radiative efficiency, and ODSs grouped as CFC12-eq similarly (fig. 4d,e). Small differences between the models in future scenarios may be a result of the assumption of a change in the rate of the Brewer-Dobson circulation in MAGICC6 (Meinshausen et al., 2011a), which increases the efficiency of the stratospheric sink for these gases. This temperature-dependent effect is not included in FAIR. Over the historical period, the differences are a result of the natural emissions of $CF_4$ (contributing to HFC134a-eq), and $CH_3Br$ and $CH_3Cl$ (contributing to CFC12-eq) providing a non-zero background state of these greenhouse gas equivalents in FAIR. In the RCP historical dataset

these background concentrations have not been added to the HFC134a-eq and CFC12-eq timeseries.

### 4.3   Historical, present and future radiative forcing

Figure 5 shows the comparison between FAIR and MAGICC6 for the 13 forcing agents considered in FAIR for the NROY ensemble. The ERF time series for the historical period in AR5 is also shown (IPCC, 2013). The updated radiative forcing relationship for $CH_4$ increases radiative forcing substantially (fig. 5b). The new relationship for $N_2O$ results in a slightly lower

ERF estimate in FAIR than RF in MAGICC6 (fig. 5c), but the ERFs from FAIR and RFs from MAGICC for $CO_2$ and the minor greenhouse gases are similar (fig. 5a,d). For non-$CO_2$ gases, AR5 did not provide a breakdown of ERF by individual gas, so the total ERF has been scaled by the ratios in the MAGICC6 timeseries.

For tropospheric ozone, the multilinear regression method based on precursor emissions very closely emulates the AR5 ERF time series and produces sensible, albeit low-biased future ERF estimates compared to MAGICC6 (fig. 5e). The shape

of the stratospheric ozone ERF curve between AR5 and MAGICC6 differs, but it can be seen that the AR5 historical ERF is well emulated (fig. 5f). Stratospheric water vapour from methane oxidation depends on the underlying methane forcing and shows the same shape as the $CH_4$ forcing curve but with better agreement to the MAGICC6 timeseries than the AR5 timeseries (fig. 5g). Contrail ERF shows a similar time evolution over the historical period to AR5 (fig. 5h). Historically, ERF from aviation contrails has been small, but may become more substantial in the future.




The median aerosol ERF in FAIR is slightly more negative than in AR5, which suggests that runs with larger negative aerosol forcing were more likely to satisfy the observed temperature constraint as the AR5 ERF time series was used to tune the emission coefficients (fig. 5i). The resulting median time series replicates the aerosol radiative forcing from MAGICC6 remarkably well from 1900 to the end of the historical period. In the future RCP scenarios, the best estimate of aerosol ERF in FAIR is stronger than in MAGICC6.

BC on snow has a smaller ERF in FAIR than the corresponding RF in MAGICC6, although the efficacy factor of 3 used in FAIR results in a similar effect on temperature between the models (fig. 5j). Estimates of future land use forcing in FAIR follow a similar shape to the Meinshausen et al. (2011b) dataset with slightly less negative best estimates to the AR5 ERF 2011 best estimate; agreement in the historical period to either MAGICC6 or AR5 is less good, but the general trajectory of forcing is correct (fig. 5k). There are substantial differences between the volcanic forcing datasets in AR5 and the RCPs that are not easy

to discern at the resolution of the plot (fig. 5l): generally, the AR5 dataset gives more negative forcing than the RCPs during volcanically active years, and also defines the absence of volcanoes as zero forcing whereas the RCP datasets define zero to be the average of the historical period, meaning quiescent years have a small positive forcing. Solar forcing is used from the new CMIP6 dataset which is reasonably similar to the RCP time series for historical forcing but exhibits some differences in the future owing to the assmued inter-cycle variability that was not present in CMIP5 (fig. 5m).

Figure 5n shows the sum of the forcing components. The best estimate sum of ERF follows AR5 closely over the historical period, which is intentional. In the RCP future scenarios, the FAIR best estimates of ERF are slightly higher than the MAGICC6 RF out to 2100 in RCP8.5, similar in RCPs 4.5 and 6.0, and lower in RCP 2.6. This is in part due to the increased $CH_4$ and contrail forcing in FAIR for RCP8.5, and more negative total aerosol ERF for RCP2.6. The FAIR model projects 2100 ERFs

(median and (5–95% credible intervals)) of 2.23 (1.46 to 3.03), 4.27 (3.40 to 5.18), 5.32 (4.16 to 6.53) and 8.71 (7.18 to 10.30) W m$^{-2}$ for RCP2.6, RCP4.5, RCP6.0 and RCP8.5 respectively.

### 4.4    Relationship between forcing components, ECS and TCR

The distribution of ERF in 2017 for aerosols, greenhouse gases and the anthropogenic total in both the FULL and the NROY ensemble assuming the RCP8.5 forcing pathway is shown in fig. 6 and table 4. The temperature constraint in NROY shifts the

distribution of ERF for greenhouse gases slightly to the left but is similar to the FULL ensemble. For aerosols, the distribution of ERF in NROY is wider than in FULL. The left-shifting of the aerosol and greenhouse gas ERF distributions in NROY results in a lower median estimate of net anthropogenic ERF (2.56 W m$^{-2}$ in 2017) for NROY.

There are negative correlations between aerosol radiative forcing and ECS/TCR (fig. 7). A large negative aerosol forcing requires a high ECS to balance and recreate realistic observed temperatures (Forest et al., 2006). Millar et al. (2015) highlighted

the necessity of anti-correlation between TCR and aerosol forcing in observational constraints. The aerosol forcing on TCR constraint is tighter than that on ECS, evidenced by the narrower mass of points in the TCR plot (fig. 7b) compared to the ECS plot (fig. 7a). A high value for TCR (greater than 2.5 K) or ECS (greater than 5 K) is only possible with a strong negative present-day aerosol forcing (more negative than $-1.0$ W m$^{-2}$).

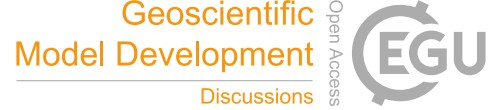



### 4.5 Observed and future temperature changes

Figure 8a shows the transient historical and RCP-projected temperature change for 1850–2100 along with the 2081–2100 median, $1\sigma$ (16–84% range) and 5–95% credible range for the NROY ensemble (fig. 8b), (c.f. Rogelj et al. (2012), Collins et al. (2013, fig. 12.8b)). For the RCP scenarios, the median NROY estimates of temperature change for 2081–2100 are 1.32, 2.12, 2.38 and 3.54 K above pre-industrial for RCPs 2.6, 4.5, 6.0 and 8.5 respectively. The median, $1\sigma$ and 5–95% ranges of total temperature change predicted from FAIR are lower than those predicted by the emissions-driven MAGICC6 experiments
(Rogelj et al., 2012; Meinshausen et al., 2009) which are reported in AR5, despite marginally higher 21st century ERF profiles in FAIR. The results of Rogelj et al. (2012) are based on best estimates of the ECS/TCR and radiative forcing from the IPCC Fourth Assessment Report (AR4), whereas in this paper we use results from AR5. Differences between this study and Rogelj et al. (2012) could be due to differences in the historical radiative forcing time series. The RF over the 1861–1880 to 2005 period, which forms the bulk of the period used to constrain the ensemble to observed temperatures, in Meinshausen et al.
(2011b) is 1.72 W m$^{-2}$ whereas the ERF differences are 1.98 W m$^{-2}$ in AR5 and 1.97 W m$^{-2}$ in FAIR (which is tuned to AR5) over the same period. Therefore, assuming Rogelj et al. (2012) used the Meinshausen et al. (2011b) radiative forcing time series or one based on AR4, the same observed temperature change would be recreated with a smaller RF in Rogelj et al. (2012) than the corresponding ERF in FAIR, and the same future forcing in MAGICC6 would lead to a higher temperature change than in FAIR. In relative terms the 2100 best estimate forcings between this study and Rogelj et al. (2012) are similar and hence the resulting future temperature predictions are lower in FAIR. Other differences between the studies include a different selection of ECS and TCR priors in Meinshausen et al. (2009) and Rogelj et al. (2012) (based on AR4, but not substantially different from the CMIP5 models used in this study), a different method of constraining to observed temperatures, and different assumptions regarding the strength of future aerosol forcing (as previously noted the median future aerosol forcing is stronger
in FAIR than in the RCPs). The sensitivity to these assumptions is tested in section 5.

### 4.6 Transient Climate Response to Emissions

There is an approximately linear relationship between cumulative $CO_2$ emissions and temperature, independent of the actual emissions pathway taken, providing temperature is still increasing (Allen et al., 2009; Collins et al., 2013). Using this linearity we can diagnose the transient climate response to emissions (TCRE), defined as the change in temperature for a 1000 Gt
cumulative emission of carbon.

We show both the TCRE assuming $CO_2$ forcing alone and the temperature change due to all forcing agents but measured against cumulative carbon emissions (fig. 9). When including the effect of non-$CO_2$ forcing on the total temperature change, the temperature response is larger than for $CO_2$ forcing alone. This indicates that a smaller cumulative $CO_2$ emission is required to reach the same temperature change, and is a result of the total non-$CO_2$ forcing being positive. This same conclusion was
reached in Collins et al. (2013) when assessing a suite of earth system models.





To determine the TCRE due to the emissions of $CO_2$ alone, we adapt the method of Tachiiri et al. (2015). Firstly it is assumed that the temperature change due to $CO_2$ alone scales with ratio of $CO_2$ forcing to total forcing:

$$T_{CO_2} = T \frac{F_{CO_2}}{F}. \tag{21}$$

Secondly, to account for the fact that the efficiency of carbon sinks are temperature-dependent and noting that $T_{CO_2} < T$ when the non-$CO_2$ forcing is positive (all times except during volcanically active periods) the cumulative emissions since 1870 are reduced by a factor that takes into account this temperature difference. We use 1870 here rather than 1850, as this is the date from which reliable estimates of carbon emissions start (Le Quéré et al., 2016) and is also at the centre of the 1861–1880 period used to evaluate temperature changes. For TCRE accounting purposes we therefore assume that cumulative carbon emissions $E_{TCRE}$ are

$$E_{TCRE} = \left( \sum_t E_{CO_2,t} \right) - \frac{r_T}{r_0 + r_C C_{acc} + r_T T} C_{acc}(T - T_{CO_2}). \tag{22}$$

The second term on the right of eq. (22) is the change in $iIRF_{100}$ due to non-$CO_2$ forcing as a fraction of the total, multiplied by the accumulated uptake. This estimates the effect on cumulative carbon emissions as a result of non-$CO_2$ forcing by the effect of non-$CO_2$ warming on decreasing efficiency of carbon sinks. The reduction factor reduces the effective cumulative emissions by around 2.5% in 2100 in RCP8.5.

The NROY ensemble in FAIR shows a TCRE of 1.03 to 2.16 K for a cumulative carbon emission of 1000 Gt with a best estimate of 1.43 K. We diagnose TCRE based on the RCP8.5 simulation but this range is almost independent of the future RCP scenario (not shown) except for RCP2.6 towards the end of the 21st century in which emissions become negative. The TCRE range from FAIR is within the range of estimates from AR5 (0.8 to 2.5 K, Collins et al. (2013)). In the original FAIR model, TCRE was shown to be 1.3 (0.8 to 2.4) K (Millar et al., 2017b). The narrowing of the credible interval between the original FAIR model and the emissions-based version is due to the results in this study being constrained to observed temperature change and the inclusion of non-$CO_2$ forcings.

Towards higher cumulative $CO_2$ emissions in RCP8.5 the temperature response has a slightly concave shape. The slight (rather than moderate) downward curvature is also present in CMIP5 earth system models, as the increase in airborne fraction of $CO_2$ with emissions almost cancels out the logarithmic relationship between $CO_2$ concentration and temperature (Millar et al., 2016).

The TCRE curve can be inverted to consider the remaining carbon budget until either 1.5 K or 2 K total warming is reached (the limits imposed in the Paris Agreement). Taking RCP8.5 as a best estimate, a total of 565 GtC has been emitted over the period 1870–2016 (Le Quéré et al., 2016). Considering all forcing agents (the red curve in fig. 9), the cumulative emissions limit required to ensure peak warming remains under 1.5 K is 790 (650–1010) GtC. To remain under 2 K of total warming, the allowable cumulative emissions are 1080 (840–1460) GtC. Millar et al. (2017a) recently showed that a further 200 GtC of additional emissions, without mitigation of non-$CO_2$ forcing agents, would result in a peak warming of less than 1.5 K in 66% of CMIP5 earth system models. We estimate here that a further 225 GtC would give a 50% chance of peak temperature rise being less than 1.5 K, which is consistent with the conclusion of Millar et al. (2017a).

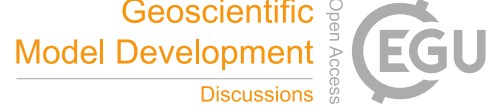

### 4.7 Top of atmosphere energy imbalance

The top of atmosphere energy imbalance $N$ can be diagnosed from (Forster et al., 2013)

$$N = F - \alpha_F T \tag{23}$$

where $\alpha_F$ is the climate feedback parameter and $\alpha_F = F_{2\times}/\text{ECS}$. In fig. 10 we compare FAIR model outputs from the NROY
ensemble under RCP4.5 to observations of the earth's energy imbalance from satellites (Clouds and the Earth's Radiant Energy
System; CERES) and from the array of Argo floats, which measure ocean temperature which is the largest component of the
change in earth's energy budget. Both datasets are taken from Johnson et al. (2016).

For every year from 2001 to 2015 the net energy balance from CERES is within the uncertainty range estimated from the
FAIR NROY ensemble. The Argo estimate of $N$ is more variable prior to 2005, after which coverage of the Argo floats saw a
large increase (Johnson et al., 2016). From 2005 onwards, all Argo estimates fall inside the credible range of FAIR estimates.
Both CERES and Argo observations from 2005 onwards are clustered towards the lower half of the credible range from FAIR,
which may indicate that ECS over the 2005–2015 period could be towards the lower end of the credible range estimated from
the NROY ensemble.

## 5 Sensitivity to prior distributions and constraints

To determine the robustness of the results of the NROY ensemble, the input assumptions were varied or the ensemble members
subjected to a different constraint as described in this section. The results are summarised in tables 5 to 7.

### 5.1 Prior distributions of ECS and TCR

The prior distributions of ECS and TCR has a large influence on the posterior distributions attained (Pueyo, 2012). Here we
test the dependence of the shape of the posterior distributions of ECS and TCR in the constrained samples on the choice of
prior distributions.

As the RWF is approximately independent of TCR we use an alternative prior starting with the distributions of TCR and
RWF. Noting the analysis of Collins et al. (2013), the AR5 likely range of TCR of 1.0 to 2.5 K is taken to be most probable,
with values between 0.5–1.0 K and 2.5–3.5 K possible but unlikely. A trapezoidal distribution in TCR with these limits is
constructed, therefore not expressing any prior judgement about the most likely value of TCR within the AR5 likely range. The
RWF is sampled from a Gaussian distribution with mean 0.6 and 5–95% range of 0.45–0.75 following Millar et al. (2017b),
truncated to fall within the 0.2–1.0 range. These ranges are subjective choices based on evidence from CMIP5 models. The
posterior distribution of ECS in particular can be sensitive to the choice of prior distribution (Frame et al., 2005; Pueyo, 2012).

Figure 11 details the alternative prior distributions and the posteriors obtained as a result of constraining to the C&W
observed temperatures. The marginal distributions of both TCR and ECS take on a lognormal shape that is similar to the
original NROY ensemble distribution (fig. 3), and weighted towards the lower end of the prior distributions. Under this prior
distribution, the modal values of ECS and TCR are about 2.1 K and 1.25 K, although the median values are somewhat higher



at 2.44 and 1.44 K, and the credible ranges are wider than in NROY (in fact, are very similar to the AR5 uncertainty ranges; table 5).

    The best estimate and credible range of ERF is very similar to NROY with the alternative prior distributions (table 6). However the future temperature projections under the RCPs span a wider range than in NROY (table 7). This is due to the wider range of ECS and TCR admitted in the posterior distributions using this alternative prior.

**5.2   ERF from a doubling of $CO_2$**

  The canonical RF value of $F_{2\times} = 3.71$ W m$^{-2}$ may not be applicable when considering all land surface and tropospheric rapid adjustments in the definition of ERF. For $CO_2$ forcing rapid adjustments include cloud changes that are not driven by temperature change (Gregory and Webb, 2008) and land surface adjustments consequential to plant stomatal conductance (Doutriaux-Boucher et al., 2009). The mean ERF for a doubling of $CO_2$ in CMIP5 models was found to be 3.44 W m$^{-2}$ (Forster

et al., 2013). The simulation is repeated with this new lower ERF value for a doubling of $CO_2$, with the same uncertainty of 20%.

    It is found that this lower value of $F_{2\times}$ does not substantially change the best estimate and credible range of ECS and TCR. The year 2100 estimates of ERF are lower than NROY as the $CO_2$ forcing that is the major component of total forcing in most years is lower. The temperature change under the RCP scenarios are higher than in NROY due to non-$CO_2$ forcings.

**5.3   Aerosol emissions-forcing relationship**

  There is evidence that the negative radiative forcing from aerosols begins to saturate as emissions increase (Twomey, 1991; Carslaw et al., 2013). In particular, Stevens (2015) used a simple model to suggest that sulfate aerosol forcing is too strong in CMIP5 models, and claims that aerosol forcing has a realistic lower bound (maximum negative) of $-1.0$ W m$^{-2}$. Subsequent analysis of the aerosol forcing and temperature change in CMIP5 models over the 1850–1950 period has questioned this conclusion (Kretzschmar et al., 2017), but we use Stevens' formulation to investigate the effects of a very different aerosol

forcing scenario. The importance of the aerosol forcing relationship is tested by using the simple relationship proposed by Stevens (2015)

$$F_{\text{aero}} = -\alpha_S E_{\text{SO}_2} - \beta_S \log\left(\frac{E_{\text{SO}_2}}{E_{\text{SO}_2,\text{nat}}}\right) \tag{24}$$

with $\alpha_S$ and $\beta_S$ forcing efficiencies and $E_{\text{SO}_2,\text{nat}}$ the background natural emissions of sulfate aerosol (note no dependence on the emissions of any other aerosol species).

To provide a distribution of inputs we follow Stevens (2015) where $1/\alpha_S$ follows a Gaussian distribution (mean 600, standard deviation 200 Tg SO$_2$ (W m$^{-2}$)$^{-1}$) and $E_{\text{SO}_2,\text{nat}}$ is Gaussian with mean 60 and 2.5–97.5% range of 30 Tg SO$_2$. $\beta_S$ is not perturbed from its default value of 0.634 W m$^{-2}$ as the uncertainty in $F_{\text{aci}}$ is captured in the variation of $E_{\text{SO}_2,\text{nat}}$. Furthermore, unlike with the linear relationship for aerosol forcing with emissions, we do not apply the AR5 uncertainty scalings to the model-derived aerosol forcing as it assumed that all of the uncertainty is captured with the variation of $\alpha_S$ and $E_{\text{SO}_2,\text{nat}}$.



The impact of this aerosol relationship is a narrowing of the credible ranges of ECS, TCR and TCRE, although with similar best estimates to NROY. The saturation effect of the aerosol ERF means there are very few ensemble members with a large negative aerosol forcing in the present day, and as such the year 2100 ERF projections are higher than NROY for all RCPs. The increase in net forcing impacts on year 2100 temperatures, which are also higher that NROY for all except the upper bound in RCP8.5.

**5.4   Historical temperature constraint**

Historical temperatures were also constrained using the HadCRUT4 dataset without infilling (Morice et al., 2012), along with the GISTEMP (Hansen et al., 2010), Berkeley Earth (Berkeley Earth, 2017) and NOAA (Zhang et al., 2017) observational datasets. The linear 1880–2016 trends are $0.91 \pm 0.18$ K, $0.99 \pm 0.22$ K, $1.07 \pm 0.16$ K and $0.93 \pm 0.24$ K respectively. All datasets, including C&W ($0.95 \pm 0.17$ K), were accessed on 17 October 2017.

We also perform analysis on the FULL dataset, where the input assumptions are guided by CMIP5 models and AR5 uncertainty ranges but no constraint to historical temperature is performed. We show in tables 5 to 7 that ECS, TCR, TCRE, ERF and temperature change depend slightly on the dataset of constraint, with datasets showing more warming over the historical period also projecting warmer 2100 temperatures under the RCP scenarios. Using the FULL ensemble however leads to wide uncertainty bounds and higher median estimates of these diagnosed parameters than using any of the constrained ensembles.

Therefore, using a historical temperature constraint rejects parameter combinations that produce larger future temperature changes.

6   **Conclusions**

We present a simple model, FAIR v1.1, that calculates global temperature change, effective radiative forcing from a variety of drivers, and concentrations of greenhouse gases. The emissions-based model is based on the FAIR v1.0 carbon cycle model

with an extension for emissions of non-$CO_2$ greenhouse gases, ozone precursors and aerosols. This version of FAIR, which is tuned to the effective radiative forcing timeseries in AR5 over the historical period, is close to the target radiative forcings from the RCP scenarios in 2100. FAIR was not tuned to emulate the radiative forcing in the MAGICC6 model, however it reproduces the trends in future radiative forcing from the RCPs in MAGICC6 and closely matches the concentrations of greenhouse gases projected.

Within FAIR, the response of the carbon cycle model can be adjusted via the rate of uptake of carbon by land and ocean processes parameterised as a function of total temperature change and cumulative carbon emissions ($iIRF_{100}$). Emissions and concentrations are converted to effective radiative forcing and the relationship of ERF to temperature change is governed by the TCR, ECS, and the efficacy of each of the 13 separate forcing categories considered in the model. The replication of specific earth system models is therefore possible as discussed by Millar et al. (2017b).

Using a correlated joint lognormal prior distribution of ECS and TCR based on CMIP5 models, running a 100,000 member ensemble in FAIR and keeping only those ensemble members that match the rate of temperature change from 1880–2016 in



from Cowtan & Way (the not ruled out yet or NROY ensemble), we find that the median and 5–95% credible range of ECS and TCR to be 2.79 (1.97 to 4.08) K and 1.47 (1.03 to 2.23) K respectively. The transient climate response to $CO_2$ emissions (TCRE) is a diagnostic output from the NROY ensemble and found to be 1.43 (1.01 to 2.16) K $(1000\,\mathrm{GtC})^{-1}$. These ranges are

similar to the IPCC AR5 likely ranges for ECS, TCR and TCRE, albeit with tighter credible bounds and shifted slightly towards the low end of the ranges. The NROY best estimates and ranges are mildly sensitive to a lower estimate of the ERF from a doubling of $CO_2$ or a different observational temperature datasets to constrain the historical temperature change rather than the Cowtan and Way (2014) dataset. They are also mildly sensitive to a non-linear aerosol forcing assumption or using a different prior ECS/TCR distribution, although the former may overconstrain ECS and TCR. All methods of constraint lead to lower

median and credible range estimates of ECS, TCR and TCRE than not constraining to temperature at all (the FULL ensemble, with input parameters estimated from the distribution of CMIP5 models and ERF uncertainties based on AR5 estimates).

Our estimate of TCR is not as low as the range derived by Otto et al. (2013) from observational constraints (0.9 to 2.0 K), although they used HadCRUT4 to constrain with the lower value of doubled-$CO_2$ ERF with a simpler regression-based model of ERF to temperature change. Similarly our best estimate of ECS is higher than the estimate provided by Gregory et al. (2016)

of around 2 K using observed sea-surface temperatures and sea-ice in two atmosphere-only GCMs. While we cannot absolutely rule out values of ECS greater than 5 K or TCR greater than 2.5 K, this would require a strong present-day aerosol forcing (at least as negative as $-1.0\,\mathrm{W\,m^{-2}}$, but probably more so) to balance. Progress towards tightening these upper bounds could therefore be achieved with a better understanding of the present-day aerosol forcing (Stevens et al., 2016).

Temperature changes projected in the NROY ensemble in 2100 are lower than those from Rogelj et al. (2012) for the RCP scenarios. This is due to the lower ensemble estimates of ECS and TCR in NROY, and the differences in present-day minus 1850 radiative forcing between AR5/NROY and the RCP radiative forcing in Meinshausen et al. (2011b). Nevertheless, under RCP8.5 the median year 2100 temperature projection is 3.94 K above the pre-industrial in our NROY ensemble, which would

have very severe global consequences. Conversely the median estimate for RCP2.6 is 1.34 K, suggesting a greater than 50% chance of limiting end-of-century warming to 1.5 K under this pathway.

The emissions-based FAIR model is useful for creating large ensembles of future temperature change based on input uncertainties in the carbon cycle parameters and effective radiative forcing strengths. This can be used for instance to assess the impacts of emissions commitment scenarios or committed warming (Ehlert and Zickfeld, 2017), or if a certain category of

emissions such as aerosols are increased or decreased in the future. FAIR can be used with integrated assessment models to calculate the social cost of carbon in the presence of non-$CO_2$ forcing agents. Following the 2015 Paris Agreement and in anticipation of the 2018 IPCC Special Report, the FAIR model can be used to investigate emissions pathways consistent with 1.5 K and 2 K total warming limits, including remaining carbon budgets, and give probabilistic indications of the likelihood of these limits being breached.

*Code availability.* The emissions-based FAIR model is available on request from the corresponding author.





*Competing interests.* The authors declare that they have no conflict of interest.

*Acknowledgements.* CJS and PMF acknowledge financial support from the Natural Environment Research Council under grant NE/N006038/1.



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



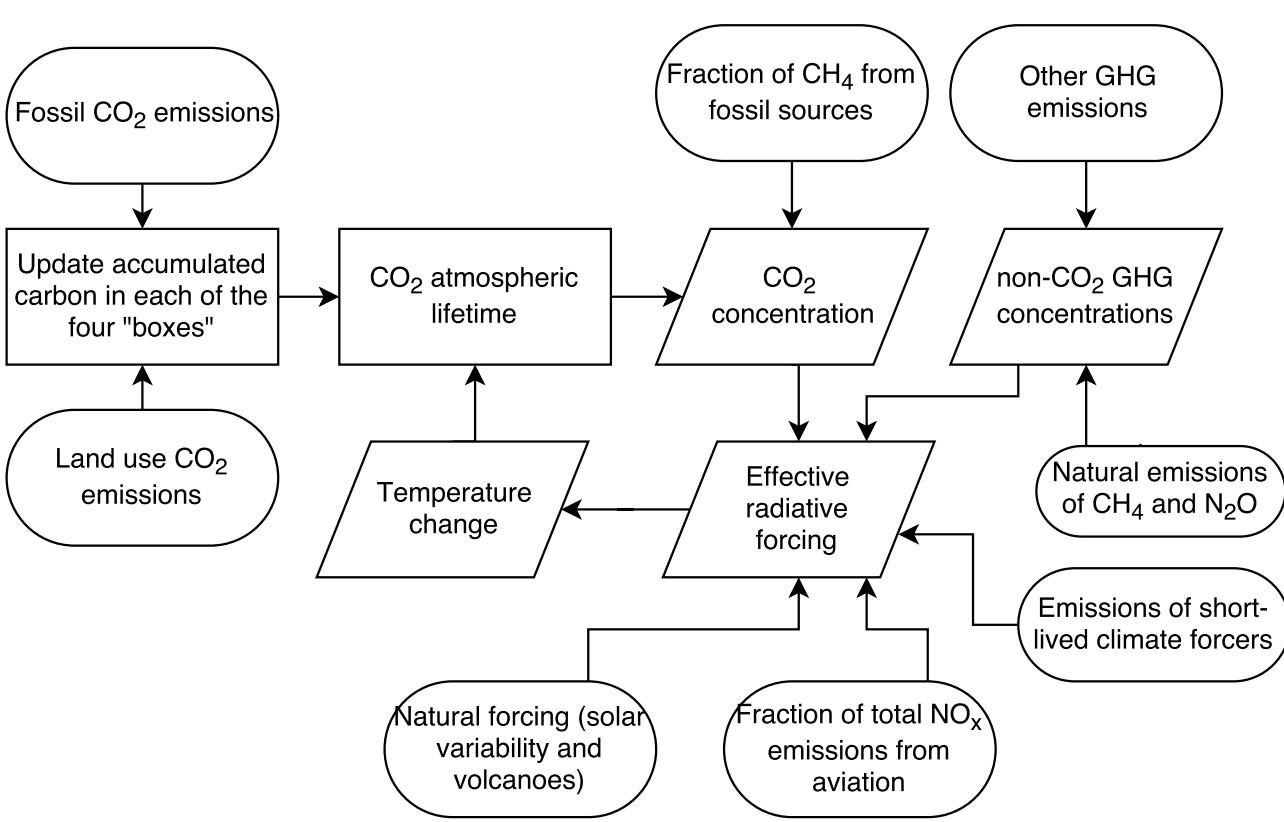

**Figure 1.** Simplified overview of the FAIR v1.1 model





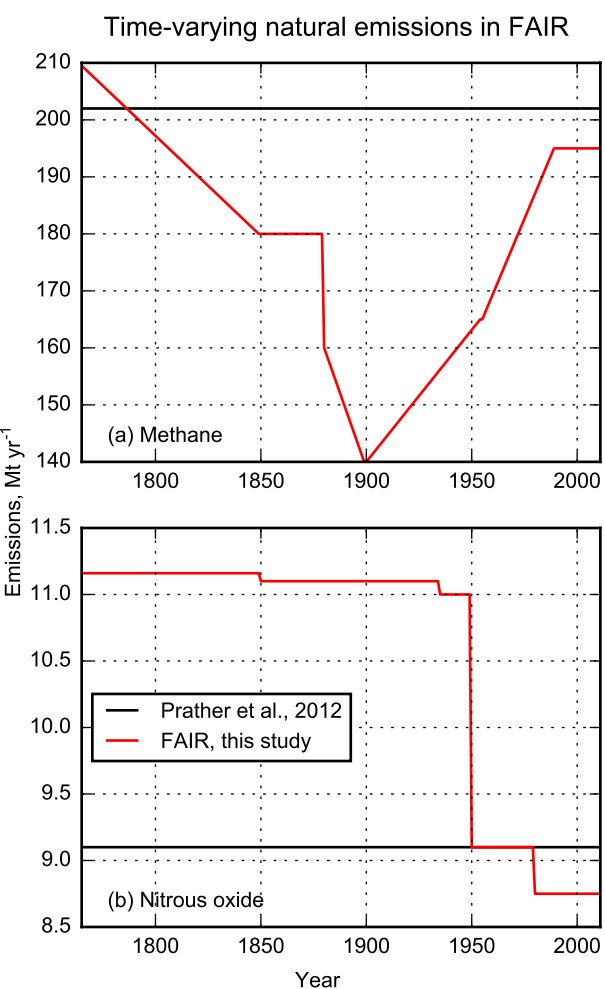

**Figure 2.** Natural emissions of methane and nitrous oxide used in the FAIR model. Future emissions are fixed at their 2011 values. Also shown are the present-day best estimates of Prather et al. (2012).



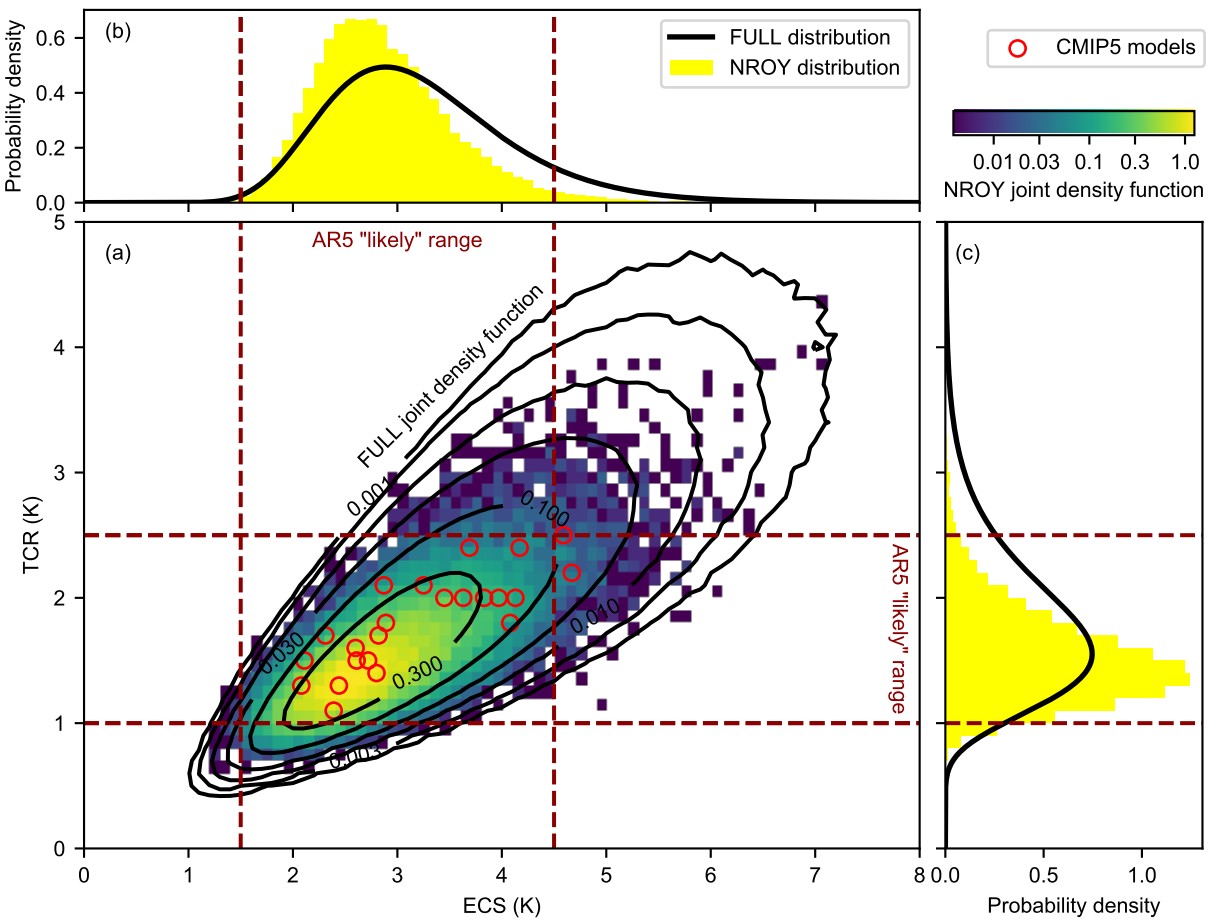

**Figure 3.** (a) Joint distributions (FULL and NROY) of ECS and TCR. (b) Marginal distributions of ECS. (c) Marginal distributions of TCR. In (a), the FULL distribution is shown with open black contours and the NROY distribution with coloured squares. Probability density is plotted on a log scale. In (b) and (c), the FULL distribution is shown as a black curve and the NROY distribution with yellow histogram bars, both plotted on a linear scale. The NROY distributions contains only those ensemble members which agree with the C&W observed historical temperatures. CMIP5 models are depicted with red circles.





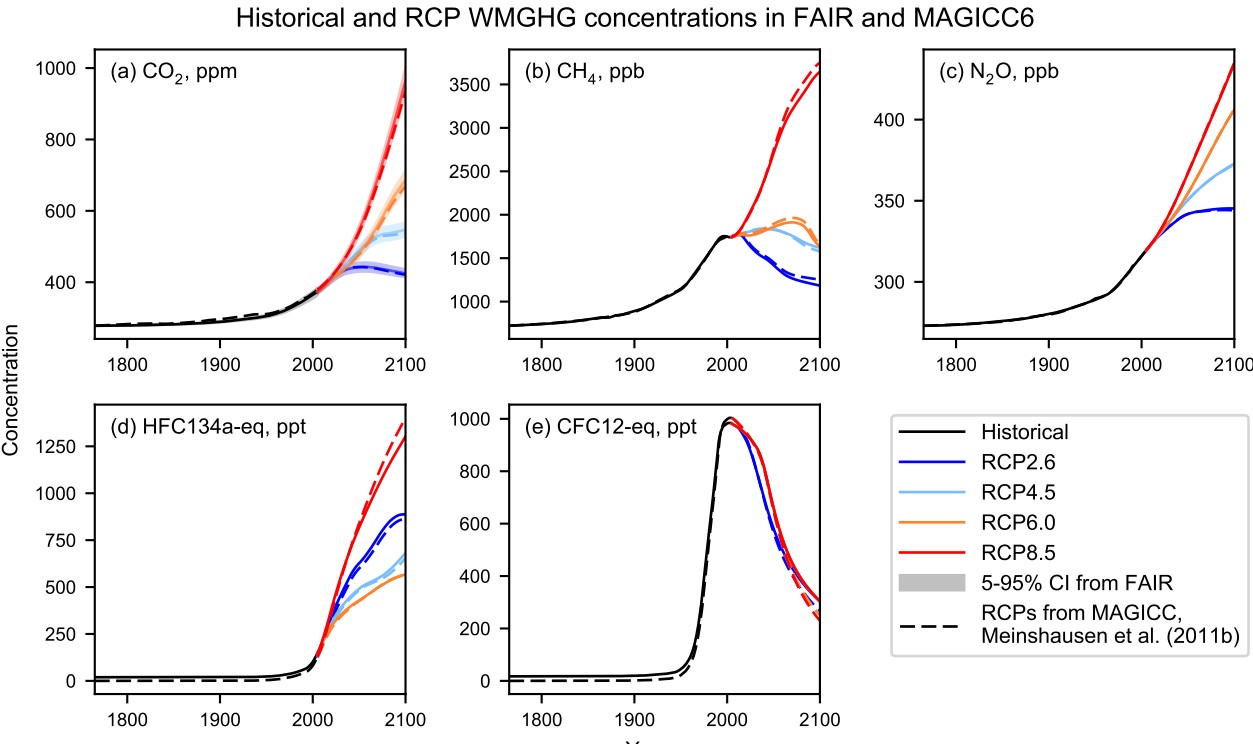

**Figure 4.** Comparison of the historical and RCP greenhouse gas concentrations in FAIR (heavy solid lines) with 5-95% confidence intervals (shading) for $CO_2$. Dashed lines show the concentrations from MAGICC6 (Meinshausen et al., 2011b). For CFC12-eq, the RCP4.5 and RCP6.0 lines lie underneath the RCP8.5 line.





**Figure 5.** Comparison of the radiative forcing from RCP2.6, RCP4.5, RCP6.0 and RCP8.5 derived from 13 separate components (subplots a–m), along with the total radiative forcing (subplot n). ERF from FAIR (solid lines) with 5-95% confidence intervals (shading), RF from MAGICC6 (dashed lines, Meinshausen et al. (2011b)) and RF from AR5 Annex II for 1850–2011 (green solid lines, IPCC (2013)).



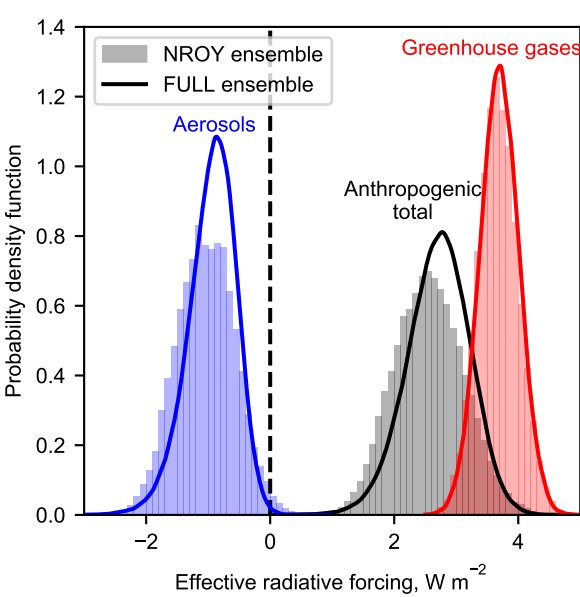

**Figure 6.** ERF from aerosols (blue), greenhouse gases (red) and total anthropogenic (black) for present-day (2017, based on RCP8.5) runs from FAIR constrained to observed temperature change (NROY; histograms) and from prior distributions (FULL; curves); compare Myhre et al. (2013b, Figure 8.16). Greenhouse gas forcing includes contributions from ozone and stratospheric water vapour from methane. Anthropogenic total is the sum of greenhouse gas, aerosol, contrails, BC on snow and land use change. The latter three distributions are not shown separately.



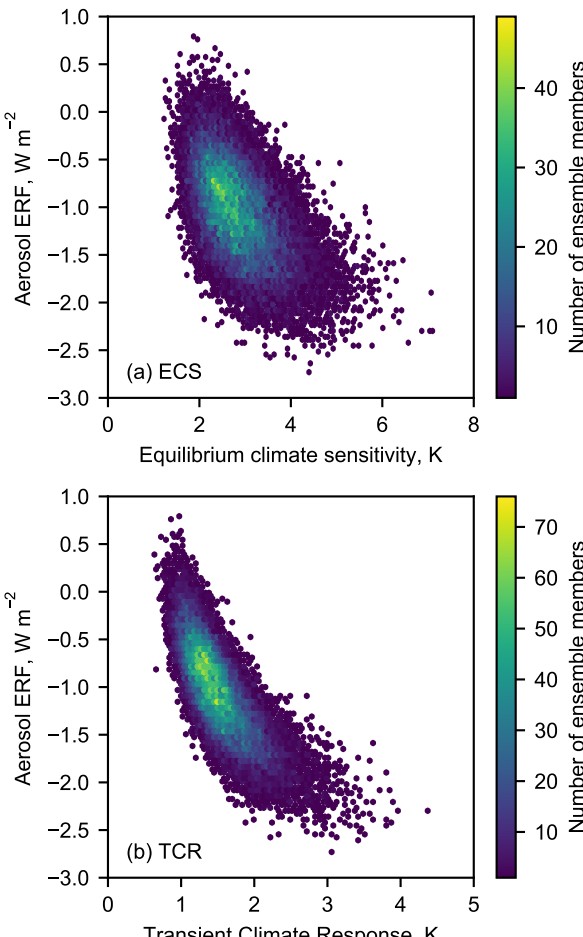

**Figure 7.** Relationship between (a) ECS and aerosol ERF; (b) TCR and aerosol ERF for the NROY ensemble. Aerosol ERF is shown for 2017 under the RCP8.5 scenario.



**Figure 8.** (a) Historical constrained modelled temperature and future probabilistic temperature scenarios from FAIR driven by emissions and forcing from the RCPs for 1850–2100 expressed as temperature change since 1861–1880. Also shown is observed temperature from C&W. (b) Comparison of 2081–2100 mean temperature from FAIR compared to the emission-driven ensemble from MAGICC6 (Rogelj et al., 2012).





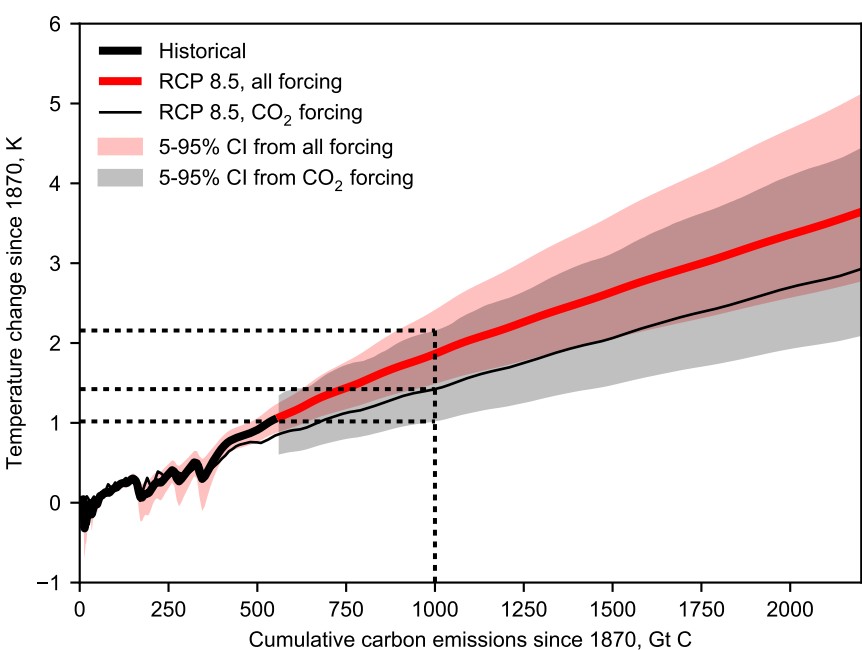

**Figure 9.** Transient climate response to $CO_2$ emissions (TCRE) for FAIR based on RCP8.5 for all forcing (red) and for $CO_2$-only forcing (black). The 90% uncertainty range for $CO_2$ forcing is not shown for historical temperature change as the ratio $F_{CO_2}/F$ is very sensitive to periods where volcanic forcing is dominant.



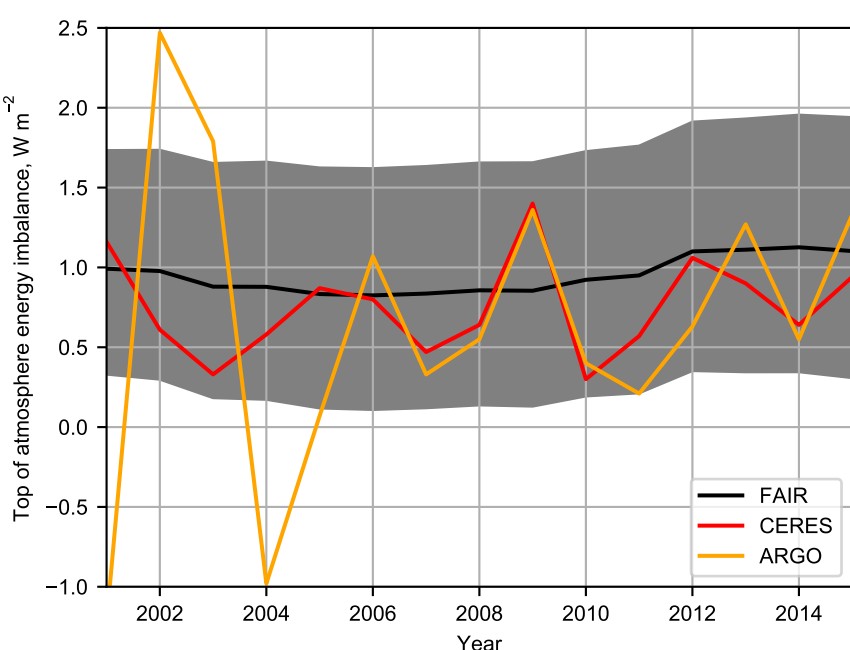

**Figure 10.** Comparison of earth's energy imbalance $N$ to observations from Argo and CERES, from Johnson et al. (2016).





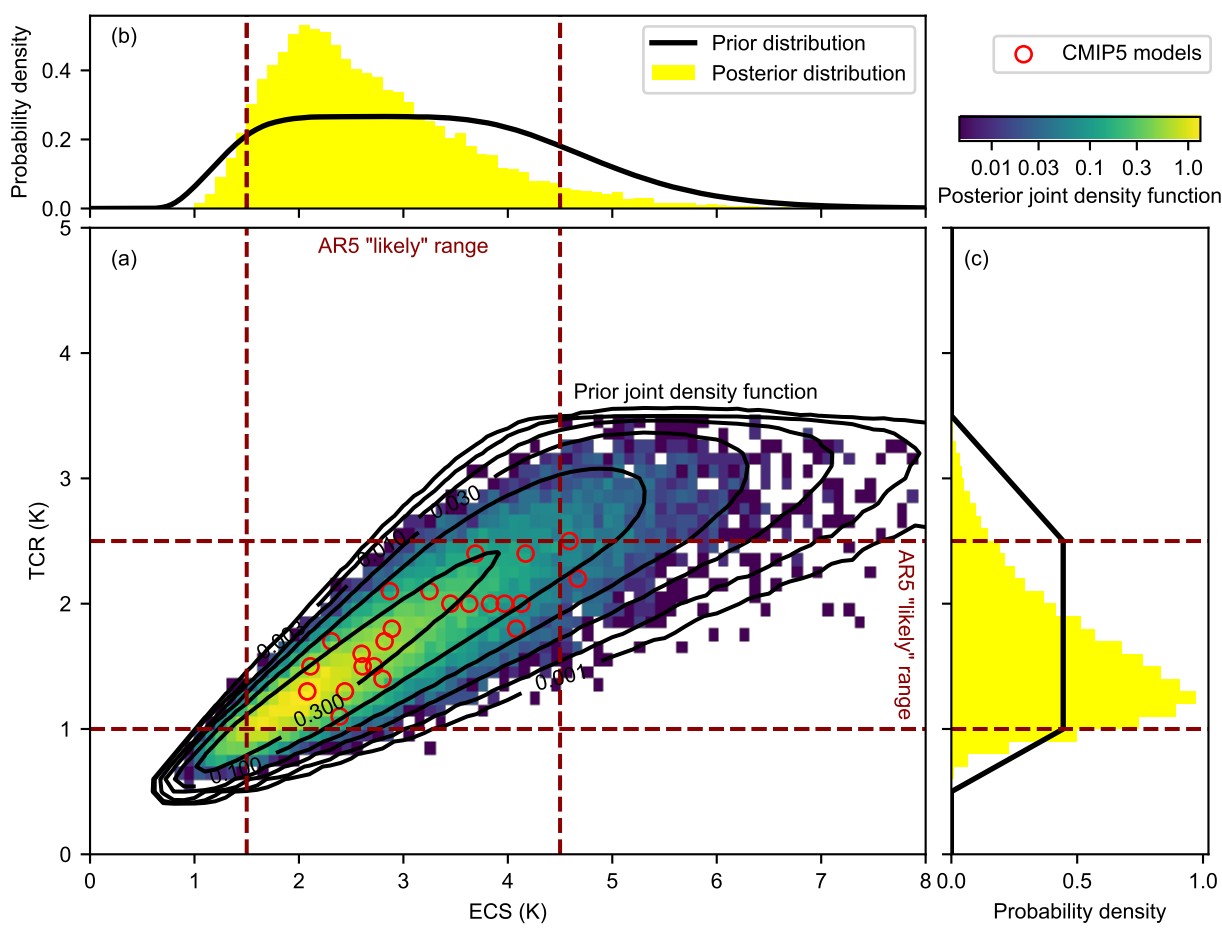

**Figure 11.** Prior and posterior distributions of the alternative ECS/TCR (a) joint distributions; (b) marginal ECS distributions; (c) marginal TCR distributions (compare fig. 3).





**Table 1.** The 13 separate forcing groups considered in FAIR v1.1 in the calculation of effective radiative forcing. The ERF uncertainty represents the 5–95% range and is used in the generation of the large ensemble (section 3). ERF uncertainties from Myhre et al. (2013b) are used except for $CH_4$ where we use the Myhre et al. (2013b) estimate inflated by the additional uncertainty in the new methane forcing relationship in Etminan et al. (2016), which also affects the uncertainty in stratospheric water vapour oxidation from methane.

| Index | Forcing agent | Depends on | ERF uncertainty |
|---|---|---|---|
| 1 | $CO_2$ | $CO_2$ emissions; $CH_4$ fossil fraction; cumulative C emissions; total temperature change | $\pm 20\%$ |
| 2 | $CH_4$ | $CH_4$ emissions | $\pm 28\%$ |
| 3 | $N_2O$ | $N_2O$ emissions | $\pm 20\%$ |
| 4 | Other greenhouse gases | Emissions of other greenhouse gases | $\pm 20\%$ |
| 5 | Tropospheric ozone | Emissions of $CH_4$ and short-lived climate forcers | $\pm 50\%$ |
| 6 | Stratospheric ozone | Concentrations of ozone depleting substances (subset of minor greenhouse gases) | $\pm 200\%$ |
| 7 | Stratospheric water vapour | $CH_4$ ERF | $\pm 72\%$ |
| 8 | Contrails | Aviation NOx fraction; total NOx emitted | $-66$ to $+191\%$ |
| 9 | Aerosols | Emissions of short-lived climate forcers | $-89$ to $+111\%$ |
| 10 | Black carbon on snow | Emissions of black carbon | $-56$ to $+128\%$ |
| 11 | Land use change | Cumulative emissions of land-use related $CO_2$ | $\pm 167\%$ |
| 12 | Volcanic | Externally supplied forcing from volcanoes | $\pm 50\%$ |
| 13 | Solar | Externally supplied forcing from solar variability | $\pm 0.05 \, \text{W m}^{-2}$ |





**Table 2.** The set of greenhouse gases used in FAIR. With the exception of methane lifetime, radiative efficiencies and lifetimes are from AR5 (Myhre et al., 2013b, table 8.A.1). For ozone-depleting substances, the fractional release coefficients $r_i$ (Daniel and Velders, 2011) and the number of chlorine and bromine atoms are also given, for calculation of equivalent effective stratospheric chlorine (eq. (13)).

| Gas | Molecular weight $w_f$ (g mol$^{-1}$) | Radiative efficiency $\eta$ (W m$^{-2}$ ppb$^{-1}$) | Lifetime $\tau$ (yr) | $r_i$ | $n_{Cl}$ | $n_{Br}$ |
|---|---|---|---|---|---|---|
| Major gases | | | | | | |
| $CO_2$ | 44.01 | N/A | Variable | | | |
| $CH_4$ | 16.04 | N/A | 9.3 | | | |
| $N_2O$ | 44.01 | N/A | 121 | | | |
| Kyoto Protocol gases | | | | | | |
| $CF_4$ | 88.00 | 0.09 | 50000 | | | |
| $C_2F_6$ | 138.01 | 0.25 | 10000 | | | |
| $C_6F_{14}$ | 338.04 | 0.44 | 3100 | | | |
| HFC23 | 70.01 | 0.18 | 222 | | | |
| HFC32 | 52.02 | 0.11 | 5.2 | | | |
| HFC43-10 | 252.06 | 0.42 | 16.1 | | | |
| HFC125 | 120.02 | 0.23 | 28.2 | | | |
| HFC134a | 102.03 | 0.16 | 13.4 | | | |
| HFC143a | 84.04 | 0.16 | 47.1 | | | |
| HFC227ea | 170.03 | 0.26 | 38.9 | | | |
| HFC245fa | 134.05 | 0.24 | 7.7 | | | |
| $SF_6$ | 146.06 | 0.57 | 3200 | | | |
| Ozone depleting substances | | | | | | |
| CFC11 | 137.37 | 0.26 | 45 | 0.47 | 3 | 0 |
| CFC12 | 120.91 | 0.32 | 100 | 0.23 | 2 | 0 |
| CFC113 | 187.38 | 0.30 | 85 | 0.29 | 3 | 0 |
| CFC114 | 170.92 | 0.31 | 190 | 0.12 | 2 | 0 |
| CFC115 | 154.47 | 0.20 | 1020 | 0.04 | 1 | 0 |
| $CCl_4$ | 153.81 | 0.17 | 26 | 0.56 | 4 | 0 |
| Methyl chloroform | 133.40 | 0.07 | 5 | 0.67 | 3 | 0 |
| HCFC22 | 86.47 | 0.21 | 11.9 | 0.13 | 1 | 0 |
| HCFC141b | 116.94 | 0.16 | 9.2 | 0.34 | 2 | 0 |
| HCFC142b | 100.49 | 0.19 | 17.2 | 0.17 | 1 | 0 |
| Halon 1211 | 165.36 | 0.29 | 16.0 | 0.62 | 1 | 1 |
| Halon 1202 | 209.82 | 0.27 | 2.9 | 0.62 | 0 | 2 |
| Halon 1301 | 148.91 | 0.30 | 65 | 0.28 | 0 | 1 |
| Halon 2401 | 259.82 | 0.30 | 20 | 0.65 | 0 | 2 |
| $CH_3Br$ | 94.94 | 0.004 | 0.8 | 0.60 | 0 | 1 |
| $CH_3Cl$ | 50.49 | 0.01 | 1 | 0.44 | 1 | 0 |





**Table 3.** Contribution to 2011 ERF from each aerosol precursor.

| Species | Allowable 2011 ERF range (W m$^{-2}$) | ERF in 2011 (W m$^{-2}$) | Regression coefficient $\gamma_i$ (eq. (15)) ($10^{-3}$ W m$^{-2}$ (Mt yr$^{-1}$)$^{-1}$) |
|---|---|---|---|
| SOx | $-1.39$ to $-0.14$ | $-0.32$ | $-5.66$ |
| BC | $0.05$ to $1.07$ | $0.11$ | $14.28$ |
| OC | $-0.02$ to $-0.61$ | $-0.32$ | $-8.90$ |
| NH$_3$ | $-0.01$ to $-0.25$ | $-0.25$ | $-5.89$ |
| NOx | $-0.01$ to $-0.10$ | $-0.10$ | $-2.62$ |



**Table 4.** Median and 5–95% credible intervals for effective radiative forcing from greenhouse gases, aerosols and anthropogenic total from the FULL and NROY FAIR ensembles in 2017. Anthropogenic total contains contributions from contrails, BC on snow and land use change and therefore is not equal to the sum of greenhouse gas and aerosol forcing. Compare fig. 6.

| | Radiative forcing (W m$^{-2}$) | |
| Forcing type | Before temperature constraint (FULL) | After temperature constraint (NROY) |
| --- | --- | --- |
| Greenhouse gases | 3.69 (3.18 to 4.21) | 3.67 (3.15 to 4.22) |
| Aerosols | −0.91 (−1.63 to −0.37) | −1.03 (−1.80 to −0.28) |
| Anthropogenic total | 2.73 (1.85 to 3.50) | 2.56 (1.66 to 3.51) |





**Table 5.** Sensitivity in the ECS, TCR, TCRE to variations in the underlying assumptions in the FAIR large ensemble. For the sensitivity experiments the section number in the manuscript describing the change is given. The Accepted column details the proportion of the 100,000 member FULL ensemble that satisfied the specified temperature constraint.

| Variation (section) | Accepted | ECS (K) | | | TCR (K) | | | TCRE (K (Eg C)$^{-1}$) | | |
|---|---|---|---|---|---|---|---|---|---|---|
| | | 5% | 50% | 95% | 5% | 50% | 95% | 5% | 50% | 95% |
| C&W temperature constraint (NROY) | 26.4% | 1.97 | 2.79 | 4.08 | 1.03 | 1.47 | 2.23 | 1.01 | 1.43 | 2.16 |
| C&W with alternative ECS/TCR prior (4.1) | 21.4% | 1.52 | 2.44 | 4.46 | 0.97 | 1.44 | 2.50 | 0.93 | 1.39 | 2.41 |
| C&W with $F_{2\times} = 3.44$ W m$^{-2}$ (4.2) | 22.1% | 1.89 | 2.71 | 4.02 | 0.97 | 1.41 | 2.19 | 1.04 | 1.48 | 2.30 |
| C&W with Stevens (2015) aerosol forcing relationship (4.3) | 34.5% | 2.09 | 2.84 | 3.92 | 1.18 | 1.51 | 1.98 | 1.12 | 1.43 | 1.84 |
| HadCRUT4 temperature constraint (4.4) | 26.2% | 1.93 | 2.74 | 4.03 | 1.00 | 1.43 | 2.20 | 0.99 | 1.39 | 2.12 |
| GISTEMP temperature constraint (4.4) | 34.1% | 1.99 | 2.82 | 4.12 | 1.05 | 1.50 | 2.27 | 1.03 | 1.46 | 2.20 |
| Berkeley Earth temperature constraint (4.4) | 25.0% | 2.07 | 2.90 | 4.18 | 1.11 | 1.56 | 2.34 | 1.11 | 1.53 | 2.27 |
| NOAA temperature constraint (4.4) | 35.2% | 1.94 | 2.77 | 4.07 | 1.01 | 1.46 | 2.23 | 0.99 | 1.42 | 2.16 |
| No temperature constraint (FULL) | 100% | 2.00 | 3.11 | 4.86 | 1.01 | 1.73 | 2.96 | 0.97 | 1.71 | 3.07 |

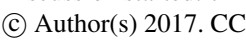



**Table 6.** Sensitivity in the effective radiative forcing to variations in the underlying assumptions in the FAIR large ensemble.

| Variation | Effective radiative forcing in 2100, W m$^{-2}$ | | | | | | | | | | | |
| | RCP 2.6 | | | RCP 4.5 | | | RCP 6.0 | | | RCP 8.5 | | |
| | 5% | 50% | 95% | 5% | 50% | 95% | 5% | 50% | 95% | 5% | 50% | 95% |
|---|---|---|---|---|---|---|---|---|---|---|---|---|
| C&W temperature constraint (NROY) | 1.46 | 2.23 | 3.03 | 3.40 | 4.27 | 5.18 | 4.16 | 5.32 | 6.53 | 7.18 | 8.71 | 10.30 |
| C&W with alternative ECS/TCR prior | 1.40 | 2.24 | 3.12 | 3.37 | 4.26 | 5.21 | 4.11 | 5.31 | 6.56 | 7.13 | 8.70 | 10.32 |
| C&W with $F_{2\times} = 3.44$ W m$^{-2}$ | 1.27 | 2.00 | 2.77 | 3.15 | 3.96 | 4.83 | 3.82 | 4.90 | 6.05 | 6.72 | 8.15 | 9.65 |
| C&W with Stevens (2015) aerosol forcing relationship | 2.35 | 2.88 | 3.44 | 3.82 | 4.61 | 5.43 | 4.91 | 5.94 | 6.99 | 7.80 | 9.29 | 10.79 |
| HadCRUT4 temperature constraint | 1.43 | 2.19 | 3.00 | 3.37 | 4.23 | 5.14 | 4.12 | 5.26 | 6.48 | 7.14 | 8.65 | 10.24 |
| GISTEMP temperature constraint | 1.48 | 2.26 | 3.07 | 3.42 | 4.29 | 5.21 | 4.18 | 5.35 | 6.56 | 7.20 | 8.75 | 10.35 |
| Berkeley Earth temperature constraint | 1.55 | 2.32 | 3.13 | 3.48 | 4.35 | 5.26 | 4.26 | 5.42 | 6.64 | 7.30 | 8.84 | 10.44 |
| NOAA temperature constraint | 1.43 | 2.21 | 3.03 | 3.37 | 4.25 | 5.17 | 4.12 | 5.29 | 6.52 | 7.14 | 8.68 | 10.28 |
| No temperature constraint (FULL) | 1.36 | 2.39 | 3.35 | 3.35 | 4.41 | 5.49 | 4.08 | 5.50 | 6.92 | 7.12 | 8.92 | 10.77 |



**Table 7.** Sensitivity in the 2100 temperature change in RCP scenarios to variations in the underlying assumptions in the FAIR large ensemble.

| Variation | temperature change in 2100 from 1861–1880 mean, K | | | | | | | | | | | |
| | RCP 2.6 | | | RCP 4.5 | | | RCP 6.0 | | | RCP 8.5 | | |
| | 5% | 50% | 95% | 5% | 50% | 95% | 5% | 50% | 95% | 5% | 50% | 95% |
|---|---|---|---|---|---|---|---|---|---|---|---|---|
| C&W temperature constraint (NROY) | 1.06 | 1.34 | 1.75 | 1.69 | 2.21 | 3.05 | 2.00 | 2.58 | 3.50 | 2.98 | 3.94 | 5.58 |
| C&W with alternative ECS/TCR prior | 0.93 | 1.25 | 1.78 | 1.48 | 2.09 | 3.33 | 1.79 | 2.46 | 3.82 | 2.69 | 3.79 | 6.20 |
| C&W with $F_{2\times} = 3.44$ W m$^{-2}$ | 1.07 | 1.37 | 1.80 | 1.74 | 2.30 | 3.23 | 2.05 | 2.67 | 3.68 | 3.08 | 4.11 | 5.98 |
| C&W with Stevens (2015) aerosol forcing relationship | 1.27 | 1.62 | 2.14 | 1.87 | 2.36 | 3.05 | 2.28 | 2.86 | 3.67 | 3.41 | 4.26 | 5.43 |
| HadCRUT4 temperature constraint | 1.01 | 1.30 | 1.70 | 1.63 | 2.14 | 2.98 | 1.92 | 2.50 | 3.41 | 2.87 | 3.82 | 5.46 |
| GISTEMP temperature constraint | 1.06 | 1.38 | 1.82 | 1.70 | 2.26 | 3.12 | 2.01 | 2.65 | 3.59 | 3.00 | 4.04 | 5.73 |
| Berkeley Earth temperature constraint | 1.16 | 1.46 | 1.88 | 1.85 | 2.37 | 3.23 | 2.20 | 2.78 | 3.71 | 3.27 | 4.23 | 5.92 |
| NOAA temperature constraint | 1.00 | 1.33 | 1.76 | 1.62 | 2.19 | 3.05 | 1.91 | 2.56 | 3.50 | 2.86 | 3.91 | 5.58 |
| No temperature constraint (FULL) | 0.87 | 1.61 | 2.94 | 1.52 | 2.63 | 4.64 | 1.75 | 3.09 | 5.51 | 2.71 | 4.73 | 8.39 |