# Peer review of "FAIR v1.2: A simple emissions-based impulse response and carbon cycle model"

_Geoscientific Model Development, 2017_

## Short Comment (SC1) · 22 Dec 2017

As explained in https://www.geoscientific-model-development.net/about/manuscript_types.html GMD is promoting that the program code (including data) described in the manuscript is publicly available through a permanent arrangement. Given the impermanence of email addresses, GMD encourages authors acting as a point of contact for obtaining the code to improve the availability with a more permanent and public arrangement. When copyright or licensing restrictions prevent the public release of model code, or in the cases where there is some other reason for not allowing public access to the code the reasons must be clearly stated in the "Code Availability" section.

Lutz Gross GMD Executive Editor

---

## Referee Comment (RC1) · W.J. Collins (Referee) · 9 Jan 2018

This paper covers two topics, the description of a simple climate model and calculation of observationally-constrained climate sensitivity. This may be doing too much in one paper. The question of climate sensitivity is long-running and of high importance, so if the authors believe they have new insight into this it would deserve its own paper with a title that reflects this and probably not in a model development journal. Conversely if the climate sensitivity calculations are intended more as an illustration of the FAIR model then much of the detail is overkill. My review will be more focussed on the model development aspects of the paper.

The FAIR model has the potential for being a very useful tool that could be widely used.

Therefore the authors need to take care that it is constructed in such a way as to be generally useful and not just for RCP scenarios. For instance it should be set up to be able to take in CO2-only emissions rather than having to subtract the non-CO2 effects from RCP8.5. The paper needs to be clear as to whether this is a model that is suitable for use by the wider community yet.

For a few of the forcing agents (e.g. aircraft, land use) there is a convoluted methodology to recreate the original activity data from the RCP emissions. A tool like this should be designed to take activity data as its basic input. It is fine for this paper if the authors have recreated the activity data from the RCP in this instance to test the model, but if the FAIR tool were to be used in an aircraft or land use study it doesn't make sense to have to generate NOx or CO2 emissions from the activity data so that FAIR can then invert them to get back to the original activity data.

It is entirely inappropriate to use the AR5 ozone and aerosol ERF time series to back out the response coefficients by linear regression. These time series were generated by a few models (it may only have been GISS) that ran forward to generate ERFs. These time series were intended to illustrate the evolution of the ERFs, not as the last word. These are not the time series that were used to force any of the CMIP5 GCMs, nor the forcings diagnosed from CMIP5 (apart maybe from GISS). Hence the ability or not to recreate the AR5 time series using FAIR is meaningless since none of the GCMs used these. Even if these time series had been more rigorously generated it is not sensible to use linear regression to derive the response coefficients as the covariances are so large. I suggest using Stevenson et al. 2013 and Aerocom to derive response coefficients. Whichever method is used, the coefficients need to be listed in tables.

Specific comments:

Page 2, line 14: Ocean sinks will become less effective too. Is this accounted for in FAIR?
[Figure]

Page 2, lines 27-30: IPCC merely used the carbon cycle responses from Joos et al. 2013 rather than constructing anything new. The Joos et al. responses were in turn taken from fits to C4MIP so would have included any feedbacks for biospheric uptake and temperature inherent in C4MIP models.

Page 3, line 7: Replace "validated" with "calibrated"

Page 3, line 11: It is not quite clear what "expected to be smoothed out in the global mean." Is trying to say. Obviously the global mean is an average of the regional variations by definition.

Page 4, equation 1: State that Ri are masses in kg.

Page 5, equation 4: State that Ct are molar mixing ratios. Equation is missing a factor delta_t.

Page 5, lines 3-5: The natural emissions in fig 2 look very unrealistic. What do MAGICC natural emissions look like? Do they have a different way of addressing this?

Page 5, lines 19-25: The methane lifetime is a function of methane concentration and this dependence is not difficult to implement, see eg MAGICC description or IPCC TAR 4.2.1.1. For increasing emissions the concentrations increase more rapidly than for a constant lifetime. This probably explains the discrepancy in the methane for RCP8.5 in fig 4(b).

Page 6, section 2.1.3: This section needs an explanation of how to avoid double counting as the CO2 emissions are often based on the total fuel consumed rather than specifically how much is fully oxidised all the way to CO2.

Page 6, line 14: Myhre et al. 2013b did not show that ERF agrees with RF, rather they found that there had not been sufficient research to determine whether the ERF was different to RF. As the authors are well aware the PDRMIP project amongst others has compared RF and ERF more recently.

Page 7, section 2.2.2: Use "well-mixed greenhouse gases" to exclude ozone.

Page 7, section 2.3.3: This linear regression is not an appropriate way to derive the response coefficients since the historical emissions strongly co-vary. Deriving a negative correspondence with NMVOC is not merely an interesting detail, it is physically wrong and so undermines the whole procedure. This must also mean that some or all of the other coefficients are overestimated to compensate. While this method may give acceptable agreement for the RCP scenarios in fig 5(e) it would give incorrect predictions when applied to more idealised scenarios e.g. if the FAIR tool were used to assess the climate impact of biomass stoves. There are sufficient data in Stevenson et al. 2013 to be able to derive more physically credible coefficients. The coefficients need to be provided in a table and compared with other studies.

Page 8, section 2.2.5. The AR5 value assumed stratospheric water vapour added 15% of the Myhre et al. 1998 methane RF. It would add a lower percentage of the Eminan et al. ERF.

Page 8, section 2.2.6: It is dangerous to build in this back calculation of aircraft activity into a tool. It is much safer to use activity data as the input. If the authors have chosen to back activity data out from RCP datasets for the purpose of this paper that's fine, but it shouldn't be hidden within the tool.

Page 8, section 2.2.7: As with ozone, linear regression is not an appropriate way to derive the response coefficients. Using speciated RFari forcing from AR5 and Aerocom to divide up the total ERFari+aci is a more transparent method. The coefficients need to be provided in a table and compared with other studies.

Page 9, section 2.2.9: Again, it is dangerous to build in this back calculation of land use activity into a tool. It is much safer to use activity data as the input. If the authors have chosen to back activity data out from RCP datasets for the purpose of this paper that's fine, but it shouldn't be hidden within the tool. The forcing is missing a minus sign.

Page 11, section 3.3: Note the +/- 20% uncertainty in the $CO_2$ ERF reflect uncertainty in our best estimate of the $CO_2$ forcing, not how it is implemented in the climate models. The actual $CO_2$ ERF "seen" by individual GCMs may lie outside this range.

Page 12, section 4.1: It is plausible that there may be an anti-correlation between a models $F_{2x}$ and its climate sensitivity (in K/(W/m2)). Is this accounted for in this study? Defining ECS and TCS in terms of $F_{2x}$ rather than in K/(W/m2) might hide some of the model variation in $F_{2x}$.

Page 13, line number 15 (actually the first line!): Given that the FAIR parameters were derived from the historical GHG concentrations, it doesn't seem much of a test that it can reproduce them.

Page 13, line number 18 (3rd line): How can MAGICC reproduce the kinks in $CO_2$, but FAIR can't?

Page 13, line number 28: The authors recognise the problems with a fixed methane lifetime. It is not difficult to implement this to rectify this errors.

Page 13, section 4.3, lines 13-14: It's not surprising the linear regression reproduces the time series it was fit to. The future ERFs need to be compared to Stevenson et al. 2013, not MAGICC.

Page 13, line 15. It is not surprising that the model can reproduce the AR5 stratospheric ozone ERF as FAIR uses exactly the same formula as AR5 (scaling with EESC).

Page 13, line 17. The reason FAIR overestimates the AR5 stratospheric water vapour value is because it scales up the Etminan et al. methane forcing which is 25% larger than the Myhre et al. 1998 forcing.

Page 15, section 4.5: Since the methane forcing is 25% stronger in FAIR, presumably the TCR has to be lower to compensate. Does this explain the lower future projections?

[Figure]

Bottom of page 15, top of page 16: I don't understand this complicated method for calculating the TCRE to CO2-alone. Surely FAIR can be forced with just CO2 emissions and will output the temperature? If this is a CO2- alone calculation why does equation 22 account for the effect of non-co2 temperature changes?

Page 18, section 5.2: This section needs to be expanded to discuss the difference between relative sensitivities in terms of F2x and absolute sensitivities in terms of K/(W/m2). If F2x is lower then the absolute sensitivity must be higher and hence the larger response when including the non-CO2 forcings.

Page 19, line numbered 18: I didn't understand why with a smaller (magnitude) present day aerosol forcing the 2100 temperatures are higher. Surely smaller aerosol forcing means lower TCR/ECS?

---

## Short Comment (SC2) · 10 Jan 2018

I am pleased to confirm that the code is now available to access from GitHub. The code pertaining to this release is available from:

https://github.com/OMS-NetZero/FAIR/archive/v1.1.tar.gz

and the project homepage is:

https://github.com/OMS-NetZero/FAIR

Alternatively the software can be installed from the Python Packages index from a Linux/GNU command line:

```
pip install fair
```

---

## Referee Comment (RC2) · B. O'Neill (Referee) · 1 Feb 2018

Overall the paper is a useful, relatively clear explanation of the FAIR 1.1 model, and its difference from FAIR 1.0, with a useful short summary of the FAIR 1.0 model. The paper also describes how model parameters are estimated by comparing outcomes to the observed temperature record, and then uses the derived parameter ranges to project future radiative forcing, concentrations, and global mean temperature change under the RCPs. This section serves to document, and exercise, the version of the model that incorporates uncertainty.

I have a few general comments followed by a number of more specific ones. First, the overall philosophy of the model could use better highlighting. It appears that choices

about design of the various model elements are guided by the desire to represent modeling approaches and parameter values presented in AR5. This is not always clear, and since other approaches are possible, it makes it a little confusing in spots why some choices were made. An early, clear statement of the approach and its rationale would be useful.

Also, as noted below, some aspects of the model are insufficiently described. Also, it would be useful to more clearly indicate when new approaches to modeling specific species are being used, and when these are borrowed from existing simple models (MAGICC or others). It would be useful, for example, to provide a summary of similarities/differences between FAIR and the MAGICC model, since MAGICC serves as a key point of reference for FAIR and for the evaluation of results.

Last, the paper seems to downplay the difference between projected warming with the FAIR model and warming according to Rogelj et al as reported in AR5. But this amounts to a full degree difference in the 2081-2100 mean under RCP8.5, which is a very substantial difference. This difference in results should be more clearly indicated (quantified in the abstract), and its reasons (difference in ECS/TCR and historical radiative forcing) pointed out more prominently. Related to this, the sensitivity of ECS/TCR (and warming) to prior distributions seems also downplayed, by indicating they are of equal importance as other factors (temperature record, eg) which have a substantially smaller quantitative effect on results. This particular factor should be identified as especially important.

Specific comments:

Abstract

The comparison of the uncertainty bounds for ECS and TCE to those reported in AR5 is worth pointing out, but should be put in context since they are not based on the same type of analysis. The AR5 range takes into account multiple lines of evidence, not just the type of study here, with a simple model constrained by observations.

The statement that the range of temperature projections under the RCP scenarios is lower in the FAIR model than those reported in AR5 is a significant outcome (especially depending on what the size of this difference actually is). The reasons for it (identified later in the paper) should also be included in the abstract.

section 2

eq 3 is not explained as clearly as it could be. in what sense is IRF-100 associated with 100 years? The description seems to indicate that it is the cumulative atmospheric carbon load over 100 years following a pulse emission, and it is being equated to an expression depending on temperature and cumulative carbon uptake, but at an unspecified date in the future. The equation should make clearer the time variable, start/end times of a 100 year period, etc. It would also be useful to give the overall intuition of this approach. I assume it is that it relates the impulse response function time constants, which are derived in conditions that do not allow for representation of dependence on sink saturation and temperature feedback, to a situation in which those processes are acting. This allows derivation of the alpha parameter representing those effects.

For methane and N2O, it seems like the approach is to specify a lifetime, and then adjust natural emissions over time so that historic concentrations are reproduced. Why is this preferable to specifying natural emissions, and estimating the lifetime that best fits the concentration data, leaving an unexplained error term that could represent variations in natural emissions or other errors (missing processes, error in anthropogenic emissions, etc.)? This is an example of where stating the general philosophy of the model might have helped, if the rationale is to use a lifetime provided in AR5. At a minimum some discussion of options and choices here is warranted.

section 2.2.3: it is unclear how well the regression approach here captures the relationship observed in data or models. Some indication of the performance of this regression model should be given, along with best estimates of coefficients.

[Figure]

2.2.4: it is noted that the functional relationship in eq 12 is from Meinshausen et al, however it is unclear if the rest of the approach (fitting to the AR5 ERF timeseries) is also the one taken by Meinshausen (or anyone else) in estimating parameters. Should be clarified what the source of the approach to the modeling and/or parameter estimation is, or whether it is new.

eq 15: as for eq 3, give some quantitative measure of how well this regression model explains the historical data (or show the scatter plot with the estimated model relationship)

2.2.9: The incorporation of the biophysical effect of land use change on forcing through albedo change may be useful, but it leaves out another important effect through changes to evapotranspiration, thus giving an incomplete accounting of biophysical effects. The authors cite one study, with one climate model, which drew conclusions based only on historical land use change, to justify including only the albedo effect. Other models will reach different conclusions about the relative effects of these two processes. Also, the effects are latitude-dependent, and the Andrews study they cite notes that the albedo effect historically has been dominated by high latitude Northern Hemisphere changes in winter (dependent on snow cover). Thus the approach of a single coefficient relating land use to albedo forcing is questionable, given that the model is intended to be applied to a wide variety of scenarios in the future with different latitudinal distributions of land use, and probably changing snow cover.

At a minimum all of these issues should be acknowledged and discussed, and the proposed approach relative to others (see eg Andy Jones paper at https://link.springer.com/article/10.1007/s10584-015-1411-5) should be justified. The quality of the approximation described by eq 17 needs to be quantified.

2.3 Temperature change: The intro to this section notes that the approach differs slightly from FAIR 1.0, but earlier in the paper FAIR 1.0 is described as a carbon cycle model. If it also includes a simple climate model, that needs to be corrected in the text.

[Figure]

section 3: In this section some of the distributions from which parameter values are drawn are specified completely, others don't seem to be. All distrbituions should be fully specified, possibly in a table, and referred to from the text.

figure 4a: it would be useful if a separate plot with different scale could be shown for the historical period. With the y axis scale set to capture the full range in 2100, it is difficult to see any detail about the relation between the FAIR range and the observations.

section 4.5: a more explicit description of the substantial differences in temperature change in 2100 between FAIR and Rogelj et al should be included. The differences in median temperature change are up to a full degree. Also, the high end of the range is very substantially truncated in FAIR, which would be extremely important to risk assessments. The text notes that they are different but underplays the size of the difference.

Table 5: section numbers referred to should all be in section 5, not 4.

Conclusions: it seems to me that the estimates of ECS and TCR (and future warming) are substantially more sensitive to the assumed priors than to other aspects of the analysis that are tested. The text here puts them all in the same category of showing "mild sensitivity". The alternative priors lead to a range of ECS/TCR that would reduce the difference in the 95% level of projected 2100 warming by half, relative to Rogelj et al. No other sensitivity would have that large of an effect.

In addition, here again the difference in projected warming by 2100 are very different from those in Rogelj et al, which seems worth emphasizing here. A full degree of warming difference in RCP8.5 is a substantial change in outlook.
* * *

---

## Author Comment (AC1) · 8 Mar 2018

Dear Brian,

Thank you for your time spent in reviewing this manuscript and the useful comments you provided. Original comments are given in bold, which are responded to point-by-point in regular font.

**General comments**

**Overall the paper is a useful, relatively clear explanation of the FAIR 1.1 model,**

**and its difference from FAIR 1.0, with a useful short summary of the FAIR 1.0 model. The paper also describes how model parameters are estimated by comparing outcomes to the observed temperature record, and then uses the derived parameter ranges to project future radiative forcing, concentrations, and global mean temperature change under the RCPs. This section serves to document, and exercise, the version of the model that incorporates uncertainty.**

Thank you for your largely positive comments overall. In addition to incorporating uncertainty there are several processes that convert emissions of non-$CO_2$ species to radiative forcing, which is a development from FAIR v1.0.

**I have a few general comments followed by a number of more specific ones. First, the overall philosophy of the model could use better highlighting. It appears that choice about design of the various model elements are guided by the desire to represent modeling approaches and parameter values presented in AR5. This is not always clear, and since other approaches are possible, it makes it a little confusing in spots why some choices were made. An early, clear statement of the approach and its rationale would be useful.**

We agree that this could have been made clearer. In the introduction we now state:

"The model philosophy in FAIR is to represent these processes as simply as possible, and to be able to emulate the ERF time series in AR5 given input emissions. FAIR is written in Python and open source."

Also, we have taken this opportunity to rewrite some of the other paragraphs in the introduction to improve readability and conciseness.

**Also, as noted below, some aspects of the model are insufficiently described. Also, it would be useful to more clearly indicate when new approaches to modeling specific species are being used, and when these are borrowed from existing simple models (MAGICC or others). It would be useful, for example, to provide a**

summary of similarities/differences between FAIR and the MAGICC model, since MAGICC serves as a key point of reference for FAIR and for the evaluation of results.

We have strived to make the treatment of each process a little clearer. We highlight in section 2.2 where a process is borrowed from MAGICC or elsewhere; where this is not stated, it has been derived by the authors.

Many of the processes in MAGICC (e.g. the carbon cycle) are not easy to summarise in a simple table, and some are not known to the authors (one example being the assumptions of natural emissions used in MAGICC), so we have not included this comparison. In further work we will investigate the different responses between the models in more detail.

**Last, the paper seems to downplay the difference between projected warming with the FAIR model and warming according to Rogelj et al as reported in AR5. But this amounts to a full degree difference in the 2081-2100 mean under RCP8.5, which is a very substantial difference. This difference in results should be more clearly indicated (quantified in the abstract), and its reasons (difference in ECS/TCR and historical radiative forcing) pointed out more prominently. Related to this, the sensitivity of ECS/TCR (and warming) to prior distributions seems also downplayed, by indicating they are of equal importance as other factors (temperature record, eg) which have a substantially smaller quantitative effect on results. This particular factor should be identified as especially important.**

Thank you for this important comment. Following the comments from the other reviewer, some of the assumptions made for the scientific components of the model have now been improved, particularly for tropospheric ozone and aerosols. The consequences are that the constrained distributions of ECS and TCR are a little higher than previously and we don't see the full degree difference in RCP8.5 any more (it is about 0.5K).

The last sentence of the abstract has been modified:

"The range of temperature projections under RCP8.5 for 2081–2100 in the constrained FAIR model ensemble is lower than the emissions-based estimate reported in AR5 by half a degree, owing to differences in forcing assumptions and ECS/TCR distributions."

**Specific comments**

**Abstract**

**The comparison of the uncertainty bounds for ECS and TCE to those reported in AR5 is worth pointing out, but should be put in context since they are not based on the same type of analysis. The AR5 range takes into account multiple lines of evidence, not just the type of study here, with a simple model constrained by observations.**

Thank you for this comment. In retrospect the AR5 figures are a "likely" ($> 66\%$) range so we were in fact not comparing the same ranges. This sentence has been updated:

"The constrained estimates of equilibrium climate sensitivity (ECS), transient climate response (TCR) and transient climate response to cumulative $CO_2$ emissions (TCRE) are 2.93 (2.04 to 4.32) K, 1.59 (1.07 to 2.50) K and 1.44 (0.97 to 2.31) K (1000 GtC)$^{-1}$ (median and 5–95% credible intervals). These are in good agreement, with tighter uncertainty bounds, than the AR5 likely range, noting that AR5 estimates were derived from a combination of climate models, observations and expert judgement."

We have also updated the commentary in section 4.1 where these results are discussed.

**The statement that the range of temperature projections under the RCP scenarios is lower in the FAIR model than those reported in AR5 is a significant out-**

**come (especially depending on what the size of this difference actually is). The reasons for it (identified later in the paper) should also be included in the abstract.**

As stated above, this has now been changed.

**section 2**

**eq 3 is not explained as clearly as it could be. in what sense is IRF-100 associated with 100 years? The description seems to indicate that it is the cumulative atmospheric carbon load over 100 years following a pulse emission, and it is being equated to an expression depending on temperature and cumulative carbon uptake, but at an unspecified date in the future. The equation should make clearer the time variable, start/end times of a 100 year period, etc. It would also be useful to give the overall intuition of this approach. I assume it is that it relates the impulse response function time constants, which are derived in conditions that do not allow for representation of dependence on sink saturation and temperature feedback, to a situation in which those processes are acting. This allows derivation of the alpha parameter representing those effects.**

You have the correct intuition for how this works. Equation 3 parametrises what would be the integrated additional carbon loading after 100 years (iIRF$_{100}$) in response to a one-time (strictly infinitesimal) pulse emission of $CO_2$ at the current point in time in the model's integration. Knowing iIRF$_{100}$ allows the scaling factor, $\alpha$, on the model carbon cycle response timescales to be calculated. We parametrise the continuous evolution of iIRF$_{100}$ in response to this purely hypothetical pulse experiment to evolve with the present climate state. Numerically, this is implemented in equation 3, by using $T$ and the cumulative carbon uptake from the previous model timestep.

**For methane and N2O, it seems like the approach is to specify a lifetime, and then adjust natural emissions over time so that historic concentrations are reproduced. Why is this preferable to specifying natural emissions, and estimat-**

**ing the lifetime that best fits the concentration data, leaving an unexplained error term that could represent variations in natural emissions or other errors (missing processes, error in anthropogenic emissions, etc.)? This is an example of where stating the general philosophy of the model might have helped, if the rationale is to use a lifetime provided in AR5. At a minimum some discussion of options and choices here is warranted.**

Note that we have now updated the natural emissions to exactly balance the concentrations given anthropogenic emissions. It is our understanding that MAGICC does something akin to the reverse, where they start with concentrations and back out natural emissions given the anthropogenic emissions.

We have experimented with a constant natural emissions rate in model development. We find that the trajectory of historical concentrations is unrealistic (especially for methane), and this problem is confounded by using time-varying atmospheric lifetimes. Additionally, natural emissions are uncertain and vary interannually. The timeseries provided in figure 2 are the model defaults, and the user can specify their own.

You are correct that as far as possible we wanted to use AR5 estimates to inform the model. For methane this was not possible; using the 12.6 year lifetime in AR5 gives future emissions that are too high, whereas using 9.3 years gives the expected results both over the historical period (for a reasonable level of natural emissions) and in future, where they agree fairly well with the RCPs. We give some justification for this at the end of section 2.1.2.

Added: "We prefer to use varying natural emissions with a fixed atmospheric lifetime of $CH_4$ and $N_2O$, firstly because this provides a good match to observed and projected concentrations and secondly because this is consistent with the simple model philosophy. Other methods of calculating concentrations of these gases are possible, for example using a fixed natural background emission and relating any differences between observed and calculated historical concentrations as an error term (either in the

natural or anthropogenic time series or missing processess), or by adjusting the atmospheric lifetime of each gas over the historical period in order to match the observed concentrations at each time step."

**section 2.2.3: it is unclear how well the regression approach here captures the relationship observed in data or models. Some indication of the performance of this regression model should be given, along with best estimates of coefficients**

As acknowledged in our response to the first reviewer, we have now moved from using a regression-based approach to one that is informed by estimates from ACCMIP models (Stevenson et al., 2013). The evolution over the historical period is similar to AR5 up to around 1970 (fig. 5e in paper), after which the ACCMIP relationship results in a slightly stronger forcing than estimated by AR5 (but well within the uncertainty range in AR5).

The main difference is in the evolution of RCP8.5 past 2005 compared to the regression-based relationship which now shows a tropospheric forcing some 0.2 W m$^{-2}$ higher than before. This is consistent with the modelling in Stevenson et al. (2013). The ozone forcing coefficients and year-2000 forcing values are provided in table 4.

**2.2.4: it is noted that the functional relationship in eq 12 is from Meinshausen et al, however it is unclear if the rest of the approach (fitting to the AR5 ERF timeseries) is also the one taken by Meinshausen (or anyone else) in estimating parameters. Should be clarified what the source of the approach to the modeling and/or parameter estimation is, or whether it is new.**

The function takes the same form as Meinshausen et al. (2011), as this was the best simple model that could be found in the literature. We do not know what the basis of their relationship is. By training our curve fit to the ERF AR5 time series we get a different parameter combination to Meinshausen et al. (2011).

We have updated this description:

"$a = -1.46 \times 10^{-5}$, $b = 2.05 \times 10^{-3}$ and $c = 1.03$ in eq. (12) are fitting parameters that are found by a least-squares curve fit between eq. (12) and the stratospheric ozone ERF timeseries from AR5; due to this data fitting approach, our parameters differ from MAGICC."

**eq 15: as for eq 3, give some quantitative measure of how well this regression model explains the historical data (or show the scatter plot with the estimated model relationship)**

As with tropospheric ozone, the aerosol forcing relationship has been updated to use established model results from Aerocom (for ERFari) and a simulator of the Ghan et al. (2013) model for ERFaci. The relationship of how each species affects ERFari (in terms of forcing per Mt emissions) and the ERFari in year 2011 is given in table 4.

The underlying model that calculates ERFaci is too slow to run in FAIR for large ensembles so was emulated based on emissions of SOx and primary organic aerosol (BC+OC). The relationship to precursor emissions and how it compares to Ghan's model is shown in figure S1.

**2.2.9: The incorporation of the biophysical effect of land use change on forcing through albedo change may be useful, but it leaves out another important effect through changes to evapotranspiration, thus giving an incomplete accounting of biophysical effects. The authors cite one study, with one climate model, which drew conclusions based only on historical land use change, to justify including only the albedo effect. Other models will reach different conclusions about the relative effects of these two processes. Also, the effects are latitude-dependent, and the Andrews study they cite notes that the albedo effect historically has been dominated by high latitude Northern Hemisphere changes in winter (dependent on snow cover). Thus the approach of a single coefficient relating land use to albedo forcing is questionable, given that the model is intended to be applied to a wide variety of scenarios in the future with different latitudinal distributions of**

**land use, and probably changing snow cover.**

**At a minimum all of these issues should be acknowledged and discussed and the proposed approach relative to others (see eg Andy Jones paper at https://link.springer.com/article/10.1007/s10584-015-1411-5) should be justified. The quality of the approximation described by eq 17 needs to be quantified.**

We appreciate our treatment of land use forcing may not include several important processes that occur in the real world that would only be possible by using an external gridded activity dataset (i.e. from LUH). However, the aim of the FAIR model is to produce a plausible projection tool with as little complexity as possible. The simpler the inputs to the model, the easier it will be for others to use it. To include gridded land processes would require something more complex than a zero dimensional model like an EMIC.

The basis for using this one coefficient was the observation that scaling with cumulative $CO_2$ land use emissions agrees remarkably well with the shape of the future forcing scenarios for the RCPs (compare dotted and solid coloured curves in fig. 5k) in MAGICC, whereas the fits to the historical data are not too bad. If MAGICC did use a more complex relationship, then it can be approximated very well with this simple formula. Although MAGICC may also contain errors and biases, we can show that the treatment of land use forcing in FAIR is no worse than in that model. We have expanded the discussion to include the points that you raise above.

The RMSE between the AR5 land use forcing and eq 17 is 0.012 W m$^{-2}$ over the historical period.

**2.3 Temperature change: The intro to this section notes that the approach differs slightly from FAIR 1.0, but earlier in the paper FAIR 1.0 is described as a carbon cycle model. If it also includes a simple climate model, that needs to be corrected in the text**

FAIR v1.0 has a temperature change component included. The 4th paragraph in section 1 is updated (now "FAIR v1.0 is well-calibrated to the carbon cycle and temperatue response of earth system models").

**section 3: In this section some of the distributions from which parameter values are drawn are specified completely, others don't seem to be. All distrbituions should be fully specified, possibly in a table, and referred to from the text.**

The ERF uncertainty ranges are given in table 1. Carbon cycle parameters are described in the text and are described as Gaussian, quoted as a mean and 90% uncertainty range. The section on ECS and TCR we have also re-written slightly and trust that it is now clearer.

For non-Gaussian ERF uncertainties the source of the original distributions are made more clear, e.g. AR5.

**figure 4a: it would be useful if a separate plot with different scale could be shown for the historical period. With the y axis scale set to capture the full range in 2100, it is difficult to see any detail about the relation between the FAIR range and the observations.**

An inset plot is now added to figure 4a which shows the historical period for $CO_2$ in more detail.

**section 4.5: a more explicit description of the substantial differences in temperature change in 2100 between FAIR and Rogelj et al should be included. The differences in median temperature change are up to a full degree. Also, the high end of the range is very substantially truncated in FAIR, which would be extremely important to risk assessments. The text notes that they are different but underplays the size of the difference.**

The new relationships we use for tropospheric ozone and aerosol forcing result in smaller temperature differences between FAIR and Rogelj et al. (2012), particularly

in the lower RCPs. For RCP8.5, FAIR is around 0.5 K lower than Rogelj et al. (2012). This is still significant and a sentence has been added: "The difference of 0.5 K in the median end-of-century warming in RCP8.5 could be particularly important in policy assessments."

We have taken this opportunity to improve the readability of section 4.5. Some super-fluous or no-longer-relevant sentences have been removed.

**Table 5: section numbers referred to should all be in section 5, not 4.**

Thank you for picking up on this reference to the old section numbering. It has now been updated.

**Conclusions: it seems to me that the estimates of ECS and TCR (and future warming) are substantially more sensitive to the assumed priors than to other aspects of the analysis that are tested. The text here puts them all in the same category of showing "mild sensitivity". The alternative priors lead to a range of ECS/TCR that would reduce the difference in the 95% level of projected 2100 warming by half, relative to Rogelj et al. No other sensitivity would have that large of an effect.**

We agree: thank you for your suggestion. We have changed the description to high-light that the ECS, TCR and TCRE posteriors are fairly insensitive to the constraining dataset whereas they are more sensitive to the prior distribution.

**In addition, here again the difference in projected warming by 2100 are very different from those in Rogelj et al, which seems worth emphasizing here. A full degree of warming difference in RCP8.5 is a substantial change in outlook.**

For the updated model, these differences are smaller (0.5K in RCP8.5), so we do not change the main description as it stands but add a few words that provides this comparison.

**References**

Ghan, S. J., Smith, S. J., Wang, M., Zhang, K., Pringle, K., Carslaw, K., Pierce, J., Bauer, S., and Adams, P. (2013). A simple model of global aerosol indirect effects. *J. Geophys. Res.-Atmos.*, 118(12):6688–6707.

Meinshausen, M., Smith, S., Calvin, K., Daniel, J., Kainuma, M., Lamarque, J.-F., Matsumoto, K., Montzka, S., Raper, S., Riahi, K., Thomson, A., Velders, G., and van Vuuren, D. (2011). The RCP Greenhouse Gas Concentrations and their Extension from 1765 to 2300. *Climatic Change*.

Rogelj, J., Meinshausen, M., and Knutti, R. (2012). Global warming under old and new scenarios using IPCC climate sensitivity range estimates. *Nat. Clim. Change*, 2:248–253.

Stevenson, D. S., Young, P. J., Naik, V., Lamarque, J.-F., Shindell, D. T., Voulgarakis, A., Skeie, R. B., Dalsoren, S. B., Myhre, G., Berntsen, T. K., Folberth, G. A., Rumbold, S. T., Collins, W. J., MacKenzie, I. A., Doherty, R. M., Zeng, G., van Noije, T. P. C., Strunk, A., Bergmann, D., Cameron-Smith, P., Plummer, D. A., Strode, S. A., Horowitz, L., Lee, Y. H., Szopa, S., Sudo, K., Nagashima, T., Josse, B., Cionni, I., Righi, M., Eyring, V., Conley, A., Bowman, K. W., Wild, O., and Archibald, A. (2013). Tropospheric ozone changes, radiative forcing and attribution to emissions in the Atmospheric Chemistry and Climate Model Intercomparison Project (ACCMIP). *Atmos. Chem. Phys.*, 13(6):3063–3085.

---

## Author Comment (AC2) · 8 Mar 2018

Dear William,

Thank you for your time taken in your detailed and mostly positive review of the manuscript. Original comments are given in bold, which are responded to point-by-point in regular font.

**General comments**

**This paper covers two topics, the description of a simple climate model and**

[Figure]

**calculation of observationally-constrained climate sensitivity. This may be doing too much in one paper. The question of climate sensitivity is long-running and of high importance, so if the authors believe they have new insight into this it would deserve its own paper with a title that reflects this and probably not in a model development journal. Conversely if the climate sensitivity calculations are intended more as an illustration of the FAIR model then much of the detail is overkill. My review will be more focussed on the model development aspects of the paper.**

Again, thank you for the time spent in reviewing the manuscript. We appreciate the point that the second part of the paper which focuses on the application of FAIR to constrain climate sensitivity may be additional detail over and above the description of the model. However, we believe it is important to show how FAIR can be useful to the climate modelling community by demonstrating its application to well-understood scenarios (the RCPs). In fact, reviewers for a paper based on this model have asked for this evidence. Since ECS and TCR must be supplied to the model, and a constraint must be applied to ensure that each combination of model parameters results in plausible output (the constraining process), it is not much of a stretch to use these results to show which input parameters of ECS and TCR result in plausible output, and provides additional confidence that the model is providing sensible results.

**The FAIR model has the potential for being a very useful tool that could be widely used. Therefore the authors need to take care that it is constructed in such a way as to be generally useful and not just for RCP scenarios. For instance it should be set up to be able to take in CO2-only emissions rather than having to subtract the non-CO2 effects from RCP8.5. The paper needs to be clear as to whether this is a model that is suitable for use by the wider community yet.**

Thank you for this positive comment. It is possible to run FAIR with $CO_2$ emissions only, where forcing from non-$CO_2$ agents can be specified optionally. For the TCRE assessment we now do this. As the $CO_2$-only treatment has already been described

in Millar et al. (2017), we do not focus on this particular implementation in the present paper.

We claim that FAIR is useful for a wide range of scenarios other than the RCPs and has been designed to evaluate a wide range of emissions commitment scenarios. As the RCPs are familiar to many and span a wide range of future projections, we use them in this paper to evaluate the performance of FAIR.

**For a few of the forcing agents (e.g. aircraft, land use) there is a convoluted methodology to recreate the original activity data from the RCP emissions. A tool like this should be designed to take activity data as its basic input. It is fine for this paper if the authors have recreated the activity data from the RCP in this instance to test the model, but if the FAIR tool were to be used in an aircraft or land use study it doesn't make sense to have to generate NOx or CO2 emissions from the activity data so that FAIR can then invert them to get back to the original activity data.**

Thank you for this important suggestion. We agree that starting from the base activity data would be more useful in order to compare impacts. However, the model is designed to be as simple as possible, and where possible to use only the 39-species emissions data appearing in the RCPs. We argue that our simple treatment of land use forcing satisfies this simple treatment while still producing a plausible time series of ERF (noting the uncertainties in land use ERF are so wide that not even the sign of the forcing is known with confidence). See also responses below and our response to the other reviewer on this point.

For the RCP scenarios, aircraft activity data does not appear to be easily available, so we do not have any data on which to train the model to. We therefore specify the (time-varying) fraction of total NOx emissions that are due to aviation as a proxy for contrail forcing. This can be calculated from the RCP databases for RCP scenarios and is included as an option in the model where the user can use a specified RCP

[Figure]

aviation NOx fraction (or provide their own). We argue that this is an improvement over MAGICC, which does not include contrail forcing (up to 0.5 W m$^{-2}$ in RCP8.5 at the 95% level). Including aircraft activity data would be a valuable future improvement.

**It is entirely inappropriate to use the AR5 ozone and aerosol ERF time series to back out the response coefficients by linear regression. These time series were generated by a few models (it may only have been GISS) that ran forward to generate ERFs. These time series were intended to illustrate the evolution of the ERFs, not as the last word. These are not the time series that were used to force any of the CMIP5 GCMs, nor the forcings diagnosed from CMIP5 (apart maybe from GISS). Hence the ability or not to recreate the AR5 time series using FAIR is meaningless since none of the GCMs used these. Even if these time series had been more rigorously generated it is not sensible to use linear regression to derive the response coefficients as the covariances are so large. I suggest using Stevenson et al. 2013 and Aerocom to derive response coefficients. Whichever method is used, the coefficients need to be listed in tables.**

This is another important point. FAIR v1.1 was designed initially to convert as best as possible raw emissions data into an ERF time series that is used to calculate temperature change for assessing a wide range of global emissions scenarios. The AR5 Annex II time series was used to correlate RCP emissions. Obviously, many forcing and emissions components do not follow linear or other simple analytical relationships, but where possible we had determined relationships that approximately track the AR5 historical ERF time series for each forcing component.

Following your comments we have implemented the ozone forcing treatment from Stevenson et al. (2013), direct aerosol forcing treatment from Aerocom (Myhre et al., 2013a), and a representation of the aerosol indirect effect from the simplified model of Ghan et al. (2013), informed by the aerosol indirect effect treatment in Stevens (2015). These changes result in different future forcing and temperature time series to v1.1, in particular a warming of around 0.5 K in RCP8.5 in 2100 compared to the old treatment

owing to the increase in tropospheric ozone and (decrease in negative) aerosol forcing in the new relationships. We feel that these changes are substantial enough to warrant an increment of the model version number, so the model version in this paper is now v1.2.

Since our modified aerosol treatment is quite similar to the model of Stevens (2015) (although with more predictor variables), we remove the Stevens aerosol relationship from the sensitivity analysis in section 5.

**Specific comments**

**Page 2, line 14: Ocean sinks will become less effective too. Is this accounted for in FAIR?**

Thank you for pointing this out – this is actually a small typo, and should read "land and ocean carbon sinks".

**Page 2, lines 27-30: IPCC merely used the carbon cycle responses from Joos et al. 2013 rather than constructing anything new. The Joos et al. responses were in turn taken from fits to C4MIP so would have included any feedbacks for biospheric uptake and temperature inherent in C4MIP models.**

We agree that the original statement was misleading, which was meant to imply that the Joos relationship as it is used in AR5 does not include any carbon or temperature feedbacks because all calculations are performed against a background concentration of 389 ppm. As you correctly suggest the original relationships in Joos et al. do include state-dependent feedbacks. We have changed "with no feedbacks assumed for biospheric carbon uptake or temperature" to "where time constants were taken from a multi-model intercomparison of full- and intermediate-complexity earth system models (Joos et al., 2013) with no feedbacks assumed for biospheric carbon uptake or

temperature".

**Page 3, line 7: Replace "validated" with "calibrated"**

We agree this is more appropriate terminology. Manuscript updated.

**Page 3, line 11: It is not quite clear what "expected to be smoothed out in the global mean." Is trying to say. Obviously the global mean is an average of the regional variations by definition.**

On reflection this sentence is superfluous and has been removed. We tried to highlight that we acknowledge that trying to account for a forcing mechanism in one global mean number may hide substantial localised variability, but this is implied from the zero-dimensional model described.

**Page 4, equation 1: State that Ri are masses in kg.**

Resolved in revised manuscript.

**Page 5, equation 4: State that Ct are molar mixing ratios. Equation is missing a factor $\delta_t$.**

$\delta_t$ is 1 (at least in this model version) so was omitted but has now been added in for completeness. $C_t$ has been explained.

**Page 5, lines 3-5: The natural emissions in fig 2 look very unrealistic. What do MAGICC natural emissions look like? Do they have a different way of addressing this?**

Originally, MAGICC natural emissions (for e.g. methane) are estimated by balancing the budget for the change in concentration, minus the contribution from anthropogenic emissions from the following relationship (see Meinshausen et al. (2011a) and the MAGICC Wiki page). The exact details do not appear to be straightforward.

We initially estimated natural emissions by adjusting them to approximate the observed

concentration time series over the RCP historical timeframe. However we have now taken a more accurate approach and formally backed out natural emissions by taking the difference in observed concentrations, accounting for atmospheric sources and sinks, and emissions, to obtain the natural contribution over the historical period. The future natural emissions are held fixed at their 2005 values. Figure 2 in the manuscript has been updated.

**Page 5, lines 19-25: The methane lifetime is a function of methane concentration and this dependence is not difficult to implement, see eg MAGICC description or IPCC TAR 4.2.1.1. For increasing emissions the concentrations increase more rapidly than for a constant lifetime. This probably explains the discrepancy in the methane for RCP8.5 in fig 4(b).**

We agree with the reviewer on this point, and we actually implemented a variable methane lifetime from Holmes et al. (2013) originally in the development process. However there are several reasons why we did not proceed in the final version:

1. The future emissions scenarios are uncertain, and a discrepancy of 5% with MAGICC for 2100 in RCP8.5 is not critical compared to the 25% error in methane forcing between Etminan et al. (2016) and Myhre et al. (1998), the latter of which we believe is used in MAGICC. Also using a sophisticated methane lifetime relationship such as that in Holmes et al. (2013) results in a deviation from MAGICC as large (in the other direction) as using a fixed lifetime for RCP8.5 in 2100.

2. There is no guarantee that the relationships that hold for small departures from present-day concentrations would apply to the more than $2\times$ increase in methane concentrations projected in MAGICC for 2100; see for example the diversity in methane lifetime and equilibrium concentrations for the NOx attribution experiment in Stevenson et al. (2013, Table 7)

3. In the context of this work, some level of natural emissions have to be assumed

post-2005 to balance the historical time series up to 2005, and it is not clear what treatment has been applied in the RCPs post-2005. To illustrate this we have experimented by implementing a simple methane feedback lifetime (where a 1% change in concentrations results in a 0.32% change in lifetime, the best estimate given in AR5), starting with an unrealistically short lifetime of 7 years in the pre-industrial in order to approximate 9.3 years in 2005, with no dependence on other effects (Holmes et al. (2013) show that methane concentration is the most important single effect on methane lifetime, even if overestimated). This results in the time series of natural balancing emissions in fig. 1 below for RCP8.5. Natural emissions of methane become negative before 2100 even under this low estimate of methane lifetime which is probably implausible; for more realistic pre-industrial and present day lifetimes, the natural emissions become more negative.

We don't rule out returning to the question of a varying methane lifetime in future model versions, and agree with the reviewer that this would be more satisfying. In our experience so far however, the additional complexity is not only unjustified but leads to unlikely methane concentrations being returned.

**Page 6, section 2.1.3: This section needs an explanation of how to avoid double counting as the $CO_2$ emissions are often based on the total fuel consumed rather than specifically how much is fully oxidised all the way to $CO_2$.**

We take the same approach as in MAGICC and do not assume that the $CO_2$ emissions inventories include oxidation of fossil sources of $CH_4$ (Meinshausen et al. (2011a, eq. A1)). The RCP scenarios, derived from MAGICC originally and also used for evaluation of FAIR, are based on total fuel consumption as detailed in Meinshausen et al. (2011b), using data from Marland et al. (2008), which in turn uses the method of Marland and Rotty (1984). Although not explicitly stated, following the argument in Marland and Rotty (1984) implies that natural gas that is not burned, incompletely combusted, or used for ammonia production accumulates as methane in the atmosphere.

Thus a molecule emitted as methane and not combusted is counted as methane until it is oxidised to $CO_2$. The atmospheric sink provides a mechanism for removal of methane. We have added the following sentence:

"As atmospheric methane concentrations are reduced by exponential decay in eq. (5), this prevents against (approximately) double-counting an emitted fossil-fuel $CH_4$ molecule as both $CH_4$ and $CO_2$ after it has been oxidised."

**Page 6, line 14: Myhre et al. 2013b did not show that ERF agrees with RF, rather they found that there had not been sufficient research to determine whether the ERF was different to RF. As the authors are well aware the PDRMIP project amongst others has compared RF and ERF more recently.**

This sentence has been changed:

"Although Etminan et al. (2016) calculate RF, Myhre et al. (2013b) concluded that over the industrial era there was not sufficient evidence to state that ERF was significantly different from RF for these three gases, and ERF is taken to equal RF, although with a doubled uncertainty range."

**Page 7, section 2.2.2: Use "well-mixed greenhouse gases" to exclude ozone.**

Included in revised version.

**Page 7, section 2.3.3: This linear regression is not an appropriate way to derive the response coefficients since the historical emissions strongly co-vary. Deriving a negative correspondence with NMVOC is not merely an interesting detail, it is physically wrong and so undermines the whole procedure. This must also mean that some or all of the other coefficients are overestimated to compensate. While this method may give acceptable agreement for the RCP scenarios in fig 5(e) it would give incorrect predictions when applied to more idealised scenarios e.g. if the FAIR tool were used to assess the climate impact of biomass stoves. There are sufficient data in Stevenson et al. 2013 to be able to derive more phys-**

**ically credible coefficients. The coefficients need to be provided in a table and compared with other studies.**

Following these comments we have re-introduced the Stevenson et al. (2013) relationships, and they are tabulated in table 4 in the manuscript. To be consistent with Stevenson et al. (2013) and the natural emissions from Skeie et al. (2011), recognising that anthropogenic emissions in 1750 were not zero, we implement a different coefficient set for RCP scenarios before 1850.

**Page 8, section 2.2.5. The AR5 value assumed stratospheric water vapour added 15% of the Myhre et al. 1998 methane RF. It would add a lower percentage of the Eminan et al. ERF.**

You are correct in that the upward revision from Etminan et al. (2016) for methane forcing would cause an upward increase in stratospheric water vapour forcing. This was detectable in fig. 5g. In v1.2, the model has been updated so that the scaling relationship depends on the forcing option used. We change the default value to 12% of the methane forcing reflecting that the ratio of Myhre et al. (1998) to Etminan et al. (2016) methane forcing is approximately 4/5. The actual scaling factor can be overridden by the user. The impact is actually fairly small at about 0.02 W m$^{-2}$ in the present day.

**Page 8, section 2.2.6: It is dangerous to build in this back calculation of aircraft activity into a tool. It is much safer to use activity data as the input. If the authors have chosen to back activity data out from RCP datasets for the purpose of this paper that's fine, but it shouldn't be hidden within the tool.**

We appreciate your concern with the treatment of contrail radiative forcing. To our knowledge, the aircraft activity figures going into the RCP datasets are not readily available so could not be used for this particular instance. Additionally, the contribution to total emissions from aviation are not available for most species; NOx is one of the few where this is available. In any case, the aviation NOx fraction is specified by the

user as a time series or constant, and the RCP values can be pre-loaded from within FAIR. We agree that an option to specify aviation activity would be a useful future development.

**Page 8, section 2.2.7: As with ozone, linear regression is not an appropriate way to derive the response coefficients. Using speciated RFari forcing from AR5 and Aerocom to divide up the total ERFari+aci is a more transparent method. The coefficients need to be provided in a table and compared with other studies.**

We agree that the old treatment may have been deficient so have updated the treatment of aerosol forcing for v1.2. Aerocom only includes the direct aerosol effect so we need a separate treatment of the indirect effect. For this we use a curve fit to the model in Ghan et al. (2013) which depends logarithmically on emissions of SOx and OC+BC. For this we have borrowed the functional dependence of aerosol forcing on SOx emissions from Stevens (2015). This is detailed in the re-written section 2.2.7 and in the supplementary material.

**Page 9, section 2.2.9: Again, it is dangerous to build in this back calculation of land use activity into a tool. It is much safer to use activity data as the input. If the authors have chosen to back activity data out from RCP datasets for the purpose of this paper that's fine, but it shouldn't be hidden within the tool. The forcing is missing a minus sign.**

The missing minus sign has been included. Thank you for spotting this omission.

We wanted to make FAIR simple, and as far as possible able to derive forcing and temperature from a single RCP-style emissions dataset. For a long time during the model development, land use was supplied externally along with solar and volcanic forcing. This was slightly unsatisfactory as land use is an anthropogenic forcing. It was noticed that land use ERF in the AR5 Annex II time series actually scaled fairly well with cumulative land-use $CO_2$ emissions, for the reasons given in the manuscript (total deforestation since pre-industrial is linked in some way to the total amount of

carbon lost to the atmosphere). Furthermore, in the future, the shapes of RCP land use forcing time series in MAGICC and FAIR are very similar, suggesting that land use $CO_2$ emissions are an important component of land use forcing in MAGICC. Our treatment is therefore no worse than in MAGICC, and probably simpler.

Requiring activity data for land use changes (from e.g. the gridded LUH dataset) would add significant complexity to the model for a small and uncertain forcing (see also response to other reviewer). However, like other suggestions, it would be a welcome development and could be implemented in a future version.

**Page 11, section 3.3: Note the +/- 20% uncertainty in the CO2 ERF reflect uncertainty in our best estimate of the $CO_2$ forcing, not how it is implemented in the climate models. The actual CO2 ERF "seen" by individual GCMs may lie outside this range.**

The 20% uncertainty applies to the 5–95% range around 3.71 W m$^{-2}$, so 10% of values going into the FULL ensemble lie outside of this range.

We should also be clear on the treatment; the Etminan et al. (2016) $CO_2$ forcing calculated by the model, from changes in GHG concentrations, is scaled to ensure that it equals the user-specified $F_{2\times}$ to ensure consistency between forcing and temperature change. A note at the end of section 2.2.1 has been added to explain this. In reality the user-specified value of $F_{2\times}$ actually makes very little difference to the temperature change (section 5), but affects the impact of non-$CO_2$ forcing on temperature as a low $F_{2\times}$ requires less non-$CO_2$ forcing to achieve the same unit temperature change.

**Page 12, section 4.1: It is plausible that there may be an anti-correlation between a models F2x and its climate sensitivity (in K/(W/m2)). Is this accounted for in this study? Defining ECS and TCS in terms of F2x rather than in K/(W/m2) might hide some of the model variation in F2x.**

No correlation was assumed between ECS and $F_{2\times}$ in the FULL ensemble. Analysing

Interactive
comment

the NROY ensemble also does not indicate any preference for high or low values of $F_{2\times}$. This is indicated in fig. 2 in this response.

Furthermore, we quickly investigated the relationship between ECS and $F_{2\times}$ from the CMIP5 results reported in Forster et al. (2013, Table 1), which shows there does not appear to be any correlation (fig. 3; this response).

**Page 13, line number 15 (actually the first line!): Given that the FAIR parameters were derived from the historical GHG concentrations, it doesn't seem much of a test that it can reproduce them.**

You are correct in that for methane and nitrous oxide, the comparisons to MAGICC6 can be made arbitrarily good by changing the natural emissions, and indeed have been in this updated model version (fig. 2 in paper), so a direct comparison over the historical period may not be relevant. The same could probably be argued for CFC12-eq and HFC134-eq, although natural emissions are only included for a few species and are not tuned to match concentrations except in the pre-industrial equilibrium case. However, concentration plots can demonstrate to potential users that the model is doing what is expected. Concentrations could also be used as an additional observational constraint in the $CO_2$ case, although we do not do that formally here as it can be seen in fig. 4a that the uncertainty around the present-day value is small in the NROY ensemble.

In the future simulations, concentration plots are useful to determine how FAIR differs from MAGICC under the simpler model setup in FAIR. It can be demonstrated the the impact of using a constant lifetime and constant natural emissions for methane and nitrous oxide is actually quite small for the RCP scenarios.

**Page 13, line number 18 (3rd line): How can MAGICC reproduce the kinks in CO2, but FAIR can't?**

The $CO_2$ time series that we compare to for pre-2005 is from observations. It is our understanding that MAGICC used prescribed GHG concentrations prior to 2005 and

emissions after this date. The historical emissions time series supplied in the RCPs are from Marland et al. (2008). By this observation we were not originally correct to *assert* that MAGICC could replicate the historical concentration time series (we don't know if it can or cannot). This has been changed in the revised version. Following a suggestion from the other reviewer we have plotted the historical $CO_2$ as an inset so the differences are easier to see.

The behaviour of $CO_2$ concentrations in the future between FAIR and MAGICC when forced with the same post-2005 emissions (fig. 4a; manuscript) shows that the carbon cycles in both models actually behave very similarly over the range of RCP scenarios in the default case. It has previously been demonstrated that MAGICC and FAIR can both be set up to emulate different carbon cycle models (see Millar et al. (2017); Meinshausen et al. (2011a)).

In reality, it is likely that there is significant inter-annual variability in the exchange between land, ocean and atmosphere that affect atmospheric $CO_2$ concentrations on yearly timescales (a recent example being the strong El Niño of 2016-17). By its design, FAIR will not capture these small deviations with four time constants. The more appropriate test is whether it can capture the long term trend between pre-industrial and present-day. Although not used formally as a constraint, $CO_2$ concentrations in the best estimates for the RCPs range between 403.4 to 408.4 ppm for 2017 in the temperature-constrained ensemble, in line with observations.

**Page 13, line number 28: The authors recognise the problems with a fixed methane lifetime. It is not difficult to implement this to rectify this errors.**

As previously discussed, using a variable methane lifetime results in implausible methane concentrations using the RCP time series. A time-varying methane lifetime would be desirable to implement, but we feel that this is not compatible with the RCP emissions data.

**Page 13, section 4.3, lines 13-14: It's not surprising the linear regression repro-**

**duces the time series it was fit to. The future ERFs need to be compared to
Stevenson et al. 2013, not MAGICC.**

This has now been done in v1.2 of the model.

**Page 13, line 15. It is not surprising that the model can reproduce the AR5 strato-
spheric ozone ERF as FAIR uses exactly the same formula as AR5 (scaling with
EESC).**

This is by design, as the model was originally intended to emulate AR5 where there
was no evidence to suggest an updated treatment (as for methane ERF). Added "as it
follows the same functional relationship as AR5".

**Page 13, line 17. The reason FAIR overestimates the AR5 stratospheric water
vapour value is because it scales up the Etminan et al. methane forcing which is
25% larger than the Myhre et al. 1998 forcing.**

This has now been updated because of the inconsistency pointed out between taking
15% of the AR5 methane forcing and 15% of the new Etminan et al. (2016) methane
forcing. The resulting time series as expected is similar to AR5.

**Page 15, section 4.5: Since the methane forcing is 25% stronger in FAIR, pre-
sumably the TCR has to be lower to compensate. Does this explain the lower
future projections?**

As you may be aware, several of the author team are involved in the IPCC Special
Report on 1.5C and this was a question that was considered. We found that the median
TCR was 4% lower with the Etminan et al. (2016) relationship. It is a contributing factor,
but not the only one. Changing the aerosol and tropospheric ozone relationships have
in fact had a bigger impact on the TCR estimates (increase of 8% between v1.1 and
v1.2).

**Bottom of page 15, top of page 16: I don't understand this complicated method
for calculating the TCRE to CO2-alone. Surely FAIR can be forced with just CO2**

**emissions and will output the temperature? If this is a CO2- alone calculation why does equation 22 account for the effect of non-co2 temperature changes?**

We agree that this was originally too complex and had been re-run in $CO_2$ only mode with much of the old description in this section deleted. Figure 9 has been updated. The TCRE does not change much, although the all-forcing (red) curve is quite a lot higher than previously, a result of an upward revision of the future non-$CO_2$ forcing.

**Page 18, section 5.2: This section needs to be expanded to discuss the difference between relative sensitivities in terms of F2x and absolute sensitivities in terms of K/(W/m2). If F2x is lower then the absolute sensitivity must be higher and hence the larger response when including the non-CO2 forcings.**

This behaviour of higher temperatures for the same forcing under a lower F2x has now been explained using the energy budget framework in eq. 21.

**Page 19, line numbered 18: I didn't understand why with a smaller (magnitude) present day aerosol forcing the 2100 temperatures are higher. Surely smaller aerosol forcing means lower TCR/ECS?**

It is actually the future evolution in aerosol ERF that affects 2100 temperature change rather than the present day aerosol forcing, which has more of a bearing on ECS and TCR. As a low present day aerosol forcing also implies a low future aerosol forcing, and the full impact of ECS and TCR on temperature change has not been realised by 2100, a low aerosol forcing implies greater temperature change over this timeframe. This is further evidenced by the switch to a new aerosol relationship which has a lower 2100 aerosol forcing than in the previous version.

Tables 5–7 (now 7–9) in the manuscript have been updated with the new values.

**References**

Etminan, M., Myhre, G., Highwood, E. J., and Shine, K. P. (2016). Radiative forcing of carbon dioxide, methane, and nitrous oxide: A significant revision of the methane radiative forcing. *Geophys. Res. Lett.*, 43(24):12,614–12,623. 2016GL071930.

Forster, P. M., Andrews, T., Good, P., Gregory, J. M., Jackson, L. S., and Zelinka, M. (2013). Evaluating adjusted forcing and model spread for historical and future scenarios in the cmip5 generation of climate models. *J. Geophys. Res.-Atmos.*, 118(3):1139–1150.

Ghan, S. J., Smith, S. J., Wang, M., Zhang, K., Pringle, K., Carslaw, K., Pierce, J., Bauer, S., and Adams, P. (2013). A simple model of global aerosol indirect effects. *J. Geophys. Res.-Atmos.*, 118(12):6688–6707.

Holmes, C. D., Prather, M. J., Søvde, O. A., and Myhre, G. (2013). Future methane, hydroxyl, and their uncertainties: key climate and emission parameters for future predictions. *Atmos. Chem. Phys.*, 13(1):285–302.

Joos, F., Roth, R., Fuglestvedt, J., Peters, G., Enting, I., von Bloh, W., Brovkin, V., Burke, E., Eby, M., Edwards, N., Friedrich, T., Frölicher, T. L., Halloran, P. R., Holden, P. B., Jones, C., Kleinen, T., Mackenzie, F. T., Matsumoto, K., Meinshausen, M., Plattner, G.-K., Reisinger, A., Segschneider, J., Shaffer, G., Steinacher, M., Strassmann, K., Tanaka, K., Timmermann, A., and Weaver, A. J. (2013). Carbon dioxide and climate impulse response functions for the computation of greenhouse gas metrics: a multi-model analysis. *Atmos. Chem. Phys.*, 13(5):2793–2825.

Marland, G., Boden, T., and Andres, R. (2008). Global, Regional, and National Fossil Fuel $CO_2$ Emissions. http://dx.doi.org/10.3334/CDIAC/00001_V2010. Accessed 12 February 2018.

Marland, G. and Rotty, R. (1984). Carbon dioxide emissions from fossil fuels: a procedure for estimation and results for 1950–1982. *Tellus B*, 36B(4):232–261.

Meinshausen, M., Raper, S., and Wigley, T. (2011a). Emulating coupled atmosphere-ocean and carbon cycle models with a simpler model, MAGICC6 – Part 1: Model description and calibration. *Atmos. Chem. Phys.*, 11:1417–1456.

Meinshausen, M., Smith, S., Calvin, K., Daniel, J., Kainuma, M., Lamarque, J.-F., Matsumoto, K., Montzka, S., Raper, S., Riahi, K., Thomson, A., Velders, G., and van Vuuren, D. (2011b). The RCP Greenhouse Gas Concentrations and their Extension from 1765 to 2300. *Climatic Change*.

Millar, R. J., Nicholls, Z. R., Friedlingstein, P., and Allen, M. R. (2017). A modified impulse-response representation of the global near-surface air temperature and atmospheric concentration response to carbon dioxide emissions. *Atmos. Chem. Phys.*, 2017:7213–7228.

Myhre, G., Highwood, E. J., Shine, K. P., and Stordal, F. (1998). New estimates of radiative forcing due to well mixed greenhouse gases. *Geophys. Res. Lett.*, 25(14):2715–2718.

Myhre, G., Samset, B. H., Schulz, M., Balkanski, Y., Bauer, S., Berntsen, T. K., Bian, H., Bellouin, N., Chin, M., Diehl, T., Easter, R. C., Feichter, J., Ghan, S. J., Hauglustaine, D., Iversen, T., Kinne, S., Kirkevåg, A., Lamarque, J.-F., Lin, G., Liu, X., Lund, M. T., Luo, G., Ma, X., van Noije, T., Penner, J. E., Rasch, P. J., Ruiz, A., Seland, Ø., Skeie, R. B., Stier, P., Takemura, T., Tsigaridis, K., Wang, P., Wang, Z., Xu, L., Yu, H., Yu, F., Yoon, J.-H., Zhang, K., Zhang, H., and Zhou, C. (2013a). Radiative forcing of the direct aerosol effect from AeroCom Phase II simulations. *Atmos. Chem. Phys.*, 13(4):1853–1877.

Myhre, G., Shindell, D., Bréon, F.-M., Collins, W., Fuglestvedt, J., Huang, J., Koch, D., Lamarque, J.-F., Lee, D., Mendoza, B., Nakajima, T., Robock, A., Stephens, G., Takemura, T., and Zhang, H. (2013b). Anthropogenic and natural radiative forcing. In Stocker, T., Qin, D., Plattner, G.-K., Tignor, M., Allen, S., Boschung, J., Nauels, A., Xia, Y., Bex, V., and Midgley, P., editors, *Climate Change 2013: The Physical Science Basis. Contribution of Working Group I to the Fifth Assessment Report of the Intergovernmental Panel on Climate Change*, pages 659–740. Cambridge University Press, Cambridge, United Kingdom and New York, NY, USA.

Skeie, R., Berntsen, T., Myhre, G., Tanaka, K., Kvalevåg, M., and Hoyle, C. (2011). Anthropogenic radiative forcing time series from pre-industrial times until 2010. *Atmospheric Chemistry and Physics*, 11(22):11827–11857.

Stevens, B. (2015). Rethinking the lower bound on aerosol radiative forcing. *J. Climate*, 28(12):4794–4819.

Stevenson, D. S., Young, P. J., Naik, V., Lamarque, J.-F., Shindell, D. T., Voulgarakis, A., Skeie, R. B., Dalsoren, S. B., Myhre, G., Berntsen, T. K., Folberth, G. A., Rumbold, S. T., Collins, W. J., MacKenzie, I. A., Doherty, R. M., Zeng, G., van Noije, T. P. C., Strunk, A., Bergmann, D., Cameron-Smith, P., Plummer, D. A., Strode, S. A., Horowitz, L., Lee, Y. H., Szopa, S., Sudo, K., Nagashima, T., Josse, B., Cionni, I., Righi, M., Eyring, V., Conley, A., Bowman, K. W., Wild, O., and Archibald, A. (2013). Tropospheric ozone changes, radiative forcing and attribution to emissions in the Atmospheric Chemistry and Climate Model Intercomparison Project (ACCMIP). *Atmos. Chem. Phys.*, 13(6):3063–3085.

[Figure]

[Figure]

**Fig. 1.** Effect on natural methane emissions for fixed and varying methane lifetime for RCP8.5

**(b)**

Probability density

**NROY distribution**

0.01 0.03 0.1 0.3
NROY joint density function

**(a)**

**(c)**

F2x (W m$^{-2}$)

ECS (K)

Probability density

**Fig. 2.** Joint and marginal histograms of ECS and F2x in the NROY ensemble (after temperature constraint)

[Figure]

**Fig. 3.** Scatter plot of ECS and F2x for CMIP5 models in Table 1 of Forster er al (2013).

---

## Author Response (AR2)

Dear Brian,

Once again thank you for the time spent in reviewing this paper and your useful suggestions. Our responses are below.

**The authors have generally done a good job in responding to reviewer comments, and I only have a couple of minor suggestions remaining.**

**The revisions to address the issue of the representation of biosphyical effects of land use are adequate, although I note that the argument was not that the FAIR model should include latitude dependence, but rather that it is unclear whether it is a good idea to represent only part of the biosphysical effects of land use. However the revised text acknowledges the other factors at play, and that the sign of the effect on forcing remains uncertain. And I would agree that a dependence on cumulative emissions from land use remains a reasonable simple relationship to rely on.**

Thank you for the understanding of our argument, which we trust is satisfactory.

Land use forcing remains an uncertainty that is difficult to relate to projections where limited information is available. As stated in the response to the other reviewer, we have now allowed the option of specifying a land-use forcing directly. The scaling with cumulative land-use $CO_2$ emissions remains available and is the default option.

Following your original suggestion in your first review, we intend to allow a gridded land-use transition dataset to be convoluted with the contributions to surface albedo change with deforestation from Jones et al. (2015) in a future model version, once this data becomes available.

**In section 3 a particular approach to constraining model parameters using the observed record of global average temperature is described. However, it needs to be made clear that while this is one approach, there is a sub-field of work on this topic, aiming to estimate model parameters (including ECS) from simple models and historical data, and there are quite a number of methodological issue to content with, including the influence of the length of the historical record, the means of distinguishing the forced response from natural variability, the method of specifying and updating prior distributions, the accounting for individual components of historical forcing, etc.**

You are quite correct. We update the paragraph at the end of section 3.1 as follows:

"It should be stressed that there are several issues to consider when attempting to derive plausible parameter sets from observational data. These include the type of observational constraints to employ (Meinshausen et al., 2009), the length of the historical record (e.g.

Otto et al. (2013)), the separation of forced response from natural variability (Haustein et al., 2017), and assumptions surrounding prior distributions (Frame et al., 2005)."

**p. 18 line 12 the revised sentence is somewhat awkwardly stated, or perhaps there is a typo. Maybe clearer to say "The RCP8.5 temperature change is lower despite higher 21st century ERF..."**

Thank you for this suggestion. We have clarified this sentence.

**References**

Frame, D. J., Booth, B. B. B., Kettleborough, J. A., Stainforth, D. A., Gregory, J. M., Collins, M., and Allen, M. R. (2005). Constraining climate forecasts: The role of prior assumptions. *Geophys. Res. Lett.*, 32(9). L09702.

Haustein, K., Allen, M., Forster, P., Otto, F., Mitchell, D., Matthews, H., and Frame, D. (2017). A real-time global warming index. *Scientific Reports*, 7(1):15417.

Jones, A. D., Calvin, K. V., Collins, W. D., and Edmonds, J. (2015). Accounting for radiative forcing from albedo change in future global land-use scenarios. *Climatic Change*, 131(4):691–703.

Meinshausen, M., Meinshausen, N., Hare, W., Raper, S. C., Frieler, K., Knutti, R., Frame, D. J., and Allen, M. R. (2009). Greenhouse-gas emission targets for limiting global warming to 2 C. *Nature*, 458(7242):1158–1162.

Otto, A., Otto, F. E. L., Boucher, O., Church, J., Hegerl, G., Forster, P. M., Gillett, N. P., Gregory, J., Johnson, G. C., Knutti, R., Lewis, N., Lohmann, U., Marotzke, J., Myhre, G., Shindell, D., Stevens, B., and Allen, M. R. (2013). Energy budget constraints on climate response. *Nat. Geosci.*, 6:415–416.

Dear William,

Once again thank you for your time spent in reviewing this manuscript and the useful comments you provided. Original comments are given in bold, which are responded to point-by-point in regular font.

Implementing the aerosol changes you suggest have changed model results slightly, so to reflect this new default behaviour we have incremented the model version to 1.3.

**Second review of Smith et al.**

**The authors have given detailed and thoughtful responses to my first review. I just have two main outstanding points.**

**As I stated before it is dangerous to hard-wire conversions from land CO2 emissions to albedo forcing, and from aircraft NOx to contrails. While these (by design) give the correct response to the RCP8.5 scenario, this model is likely to be taken up and (the authors should hope) used by many diverse groups for different purposes and different scenarios. It is very likely that this hard-wiring will persist out in the community even when the authors have updated their own version. As an example the RCP6.0 scenario has strong negative land CO2 emissions which presumably will give incorrect land albedo forcing. The SSP2.6 and 1.9 also have negative land emissions. If the input data is land use area then at least any future user is forced to think about where this data is coming from.**

Thank you for this comment.

We use this model for carbon budget calculations for the 600+ scenarios that are used in the IPCC Special Report on 1.5 Degrees. For these scenarios all that are provided are annual global mean emissions of the 39 species mentioned in table 1 in the manuscript. We therefore do not have any better data on which to calculate forcing due to land use change or aviation contrails in these scenarios. However, we have expanded the options available for both forcing categories to allow the user to select a more appropriate option.

**Land use**

We have now added an option to allow the user to specify their own time series of land use ERF externally, thereby not calculating it through a scaling with $CO_2$ land use emissions. The scaling with cumulative land-use $CO_2$ emissions remains the default option.

A further option we will develop is to use an external dataset of annual land-use transitions to and from forest/shrubland from the LUH datasets, and multiply these by the effect on albedo change in each grid cell calculated from the GCAM model in Jones et al. (2015),

figure 1. The Jones et al. (2015) data is currently not available to us, but this feature will be implemented when it is.

**Contrails**

We understand your original concern and therefore provide two alternative methods.

As for land use, a user can specify a time series of contrail forcing directly, bypassing any internally-calculated routines.

Alternatively, we provide a relationship based on a user-supplied time series of kerosene jet-fuel consumption. For RCP scenarios this data does not appear to be available so cannot be used for our demonstration of future projections, but a user could implement this if they did have the data.

**To respond to the author's responses on methane lifetime:**

**1. If using the Holmes et al. parameters gives a slightly different result to MAGICC (see also their Fig 7) that doesn't seem an issue as there is no reason to assume MAGICC is the "truth". I agree with RCP8.5 there are two opposing effects – methane increases the lifetime (+29%), climate decreases the lifetime (-17%), but that might not be the same in all scenarios e.g. where climate increases but methane doesn't or vice-versa.**

**2. The relationship has been shown (in theory, and in models) to hold for methane doubling in many papers, particularly by Michael Prather and Chris Holmes.**

**3. Applying a simple methane feedback won't work so it is not surprising it doesn't reproduce the historical or future without extreme natural emissions. Over the historical the methane self-feedback effect will be offset by increases in NOx emissions and changes in climate, Naik et al. 2013 ACP find these effects more or less cancel. In the future, as above Holmes et al. find climate change offsets more than half of the methane self-feedback in RCP8.5.**

Not using a state-varying methane lifetime remains a frustration and is an avenue for future development.

**Specific comments:**

**Page 2, line 13: Do you really mean different responses to "forcings" here (i.e. different efficacies) or do you mean different responses to emissions?**

We meant in response to emissions, and have updated this sentence.

**Page 2, line 15: Gasser et al. 2017 "Accounting for the climate–carbon feedback in emission metrics" Earth Syst. Dynam. have quantified the effect of non-CO2 agents on the carbon cycle.**

This reference has been added, and a recent paper on the effects of non-$CO_2$ forcing on carbon budgets (Tokarska et al., 2018) has also been cited.

**Page 2, line 23: I should have been clearer in my comment on this the first time round, the Joos parameter certainly do include feedbacks for both biospheric carbon uptake and temperature (see AR5 section 8.7.1.4 and above Gasser paper).**

Thank you for this clarification. We have tried to simplify further to indicate that the issue is that the Joos et al. (2013) relationship, which is designed for a particular experiment (100 GtC pulse against a background of 389ppm and present-day conditions) does not represent C4MIP carbon cycles well for a more general background state. This was the carbon-cycle relationship introduced in Millar et al. (2017). As Joos et al. (2013) models the multi-model carbon cycle response, it does include carbon-cycle-climate feedbacks.

"A simple emulation of the carbon cycle of full- and intermediate-complexity earth system models was developed by Joos et al. (2013) and used in the IPCC Fifth Assessment Report (AR5) for the purposes of calculating global warming potentials. The model was developed for a 100 GtC pulse against a background $CO_2$ concentration of 389 ppm. Millar et al. (2017) showed that this model does not sufficiently capture the time-evolving dependency of carbon sinks against different background conditions."

**Page 3, line 16: It is not obvious whether the methane is already accounted for in the carbon budgets (Boucher et al. 2009). See my further comment on page 6.**

We agree it is not obvious. From Gillenwater (2008) and Boucher et al. (2009) it appears that our treatment of including oxidised methane as $CO_2$ is correct, and MAGICC also includes a proportion of methane oxidised to $CO_2$ (eq. A1 of (Meinshausen et al., 2011)). The assumption that methane oxidised to $CO_2$ is not included in national emissions inventories has been moved to this section.

Gillenwater (2008) suggests that this treatment adds between 0.5 and 0.7% to global emissions on a GWP basis. So, neglecting this effect (or erroneously including it) does not introduce large errors.

The fraction of fossil methane oxidised to $CO_2$ is a user-specified input which defaults to 0.61 following Boucher. The time series of total methane emissions attributable to fossil sources is scenario-dependent, and RCP-calculated values are included in FAIR for the user to import. By setting either of these numbers to zero, the user can switch off methane oxidation to $CO_2$. This has been highlighted in section 2.1.3.

We have tested this in RCP4.5, and found that the difference in $CO_2$ concentrations with the methane conversion switched off is 0.3 ppm in 2015 and about 1.2 ppm in 2500.

**Page 3, line 28 & page 4, line 23: Either use "molar mixing ratio" instead of "concentration", or specifically state that the units are mol mol-1. While it is generally fine to use "concentration" colloquially, here specific equations are being presented and the reader needs to know whether the results are in kg m-3 or mol mol-1.**

Thank you for this suggestion which has been implemented.

**Page 5, lines 19-25. If you are going to mention the perturbation lifetime, I think you should explicitly say why it is not appropriate (changes in emissions of NOx & VOCs, changes in climate). Naik et al. 2013 ACP find the methane lifetime is essentially unchanged over the historical period due to compensating terms (models range from -15% to +12%).**

Thank you for this suggestion. The following commentary has been added:

"As emissions of OH-affecting species (NOx, NMVOCs, CO) and temperature have varied substantially over the historical period, the perturbation lifetime definition is not appropriate as the background state is not constant, and furthermore there is no robust evidence of a change in methane lifetime between pre-industrial and present-day from the ACCMIP inter-model comparison (Naik et al., 2013)."

**Page 6, lines 1-2. The exponential decay of methane has no affect at all on the double counting as this decay is to CO2. Even if the decay was instantaneous each mole of CH4 would end up as a mole of CO2.**

We have deleted this sentence to prevent confusion, and have mentioned the fact that methane oxidation fraction and proportion of methane from fossil origin are user-specifiable.

**Page 8, line 29-30: The semi-direct effect applies (almost) entirely to BC in IPCC. No significant semi-direct effect has been quantified for the scattering aerosols. Therefore your BC coefficient should be scaled down (and certainly not up by 1.37) until you get a forcing of -0.45, and all other coefficients unchanged.**

Thank you for this suggestion. As with many numbers supplied in this paper, the aerosol radiative efficiencies for each species are inputs to the model that can be changed by the user, and we report the defaults. Therefore we have updated these default coefficients.

The new relationship uses the 1750-2010 direct radiative forcings from Myhre et al. (2013) for non-BC aerosols and scales the direct forcing from BC down by 0.1 W m$^{-2}$ to account for the semi-direct effect to match the best estimate ERFari. Table 5 and the aerosol section in the text have been updated.

Because this change in default aerosol treatment also changes the results, the headline results reported in the paper have also changed slightly. To highlight the fact that this is a different (default) model setup (although scientifically the same model since the last paper revision), we decided it was cleaner to increment the model version to 1.3.

**Section 2.2.9: It is very dangerous to have a conversion from CO2 emissions to land-use hard-wired into FAIR – for instance in RCP6.0 land use CO2 emissions are strongly negative (I don't know why). It would be much safer to have land use area as an input. RCP data can then be pre-processed to create a land use area input. As the authors note the latitudinal dependence is very large. The new SSPs provide land use area change for different cover types.**

As noted earlier in our response, we have now provided an alternative method to directly specify land use forcing in FAIR. We have requested the gridded deforestation-to-forcing dataset from Andy Jones (but we do not yet have this), which would allow gridded land-use datasets to be imported directly.

The SSPs do provide land use area change, but they do not appear to be gridded (at least from the SSP database at `https://tntcat.iiasa.ac.at/SspDb`). Land use (presumably MAGICC-derived) forcing data does not appear to be available from these SSP scenarios either, but even if it was, this would only be tuning responses to MAGICC.

[revised manuscript text omitted]